# Inequality of household consumption and air pollution-related deaths in China

Hongyan Zhao[1,2,10], Guannan Geng [2,10], Qiang Zhang[1], Steven J. Davis[1,3,4], Xin Li [5], Yang Liu[1], Liqun Peng[2], Meng Li[1], Bo Zheng [2], Hong Huo[6], Lin Zhang [7], Daven K. Henze [8], Zhifu Mi [9], Zhu Liu [1], Dabo Guan [1] & Kebin He[2]

Substantial quantities of air pollution and related health impacts are ultimately attributable to household consumption. However, how consumption pattern affects air pollution impacts remains unclear. Here we show, of the 1.08 (0.74–1.42) million premature deaths due to anthropogenic $PM_{2.5}$ exposure in China in 2012, 20% are related to household direct emissions through fuel use and 24% are related to household indirect emissions embodied in consumption of goods and services. Income is strongly associated with air pollution-related deaths for urban residents in which health impacts are dominated by indirect emissions. Despite a larger and wealthier urban population, the number of deaths related to rural consumption is higher than that related to urban consumption, largely due to direct emissions from solid fuel combustion in rural China. Our results provide quantitative insight to consumption-based accounting of air pollution and related deaths and may inform more effective and equitable clean air policies in China.

[1] Ministry of Education Key Laboratory for Earth System Modeling, Department of Earth System Science, Tsinghua University, Beijing 100084, China. [2] State Key Joint Laboratory of Environmental Simulation and Pollution Control, School of Environment, Tsinghua University, Beijing 100084, China. [3] Department of Earth System Science, University of California, Irvine, CA 92697, USA. [4] Department of Civil and Environmental Engineering, University of California, Irvine, CA 92697, USA. [5] Department of Environmental Science and Engineering, Beijing Technology and Business University, Beijing 100048, China. [6] Institute of Energy, Environment and Economy, Tsinghua University, Beijing 100084, China. [7] Laboratory for Climate and Ocean-Atmosphere Studies, Department of Atmospheric and Oceanic Sciences, School of Physics, Peking University, Beijing 100871, China. [8] Department of Mechanical Engineering, University of Colorado Boulder, Boulder, CO 80309, USA. [9] The Bartlett School of Construction and Project Management, University College London, London WC1E 7HB, UK. [10] These authors contributed equally: Hongyan Zhao, Guannan Geng. Correspondence and requests for materials should be addressed to Q.Z. (email: qiangzhang@tsinghua.edu.cn)

Outdoor air pollution in China has caused more than 1 million premature deaths per year in recent years[1,2], and considerable research effort has thus been devoted to identifying its sources[3–7] and cost-effective mitigation options[8–11]. It is known that direct emissions from households (i.e., fuel combustion for home cooking, and/or independent heating) substantially contribute to the PM$_{2.5}$ pollution-related premature deaths in China[3,4,12–14]. Yet household consumption may also indirectly impact human health via air pollution virtually embodied in goods and services consumed[7,15–17], and regional clean air policies which focus on direct sources may thus encourage leakage of polluting activities to other regions which may have less resources to control emissions and provide health services[18]. Whereas previous studies have examined the embodied water use[19,20], energy consumption[21,22], and emissions[23–26] of regions' household consumption (including the roles of income, geography, culture, age, household size, regional policies[27–29]), no previous studies have quantified air pollution-related deaths embodied in household consumption. In addition to linking the locations of consumed goods to sources of air pollution, such a consumption-based accounting of air pollution deaths also requires tracking the physical transport of that pollution in the atmosphere and estimating the related deaths. Herein, we distinguish air pollution deaths related to emissions directly produced by a household from those related to emissions embodied in goods consumed by a household as direct and indirect, respectively.

Using a combination of four economic and physical models and province-level income and consumption statistics, we quantify the air pollution health impacts from both direct and indirect emissions of household consumption for 12 income groups (5 for rural and 7 for urban) over 30 provinces in mainland China. We use a detailed inventory for anthropogenic emissions in China (MEIC: http://www.meicmodel.org/), household consumption statistics, and a multi-regional input–output model of the Chinese economy to quantify direct and indirect pollutant emissions from household daily consumption by various income groups, and we then identify and isolate the contributions of these emissions to outdoor PM$_{2.5}$-related premature deaths by using the GEOS-Chem adjoint model combined with the integrated exposure-response (IER) concentration-response relationships[6,30,31]. Our work provide quantitative estimates of air pollution and related deaths attributed to household consumption from a consumption-based perspective and reveal their differences according to income levels and locations of the residents. The findings of this study provide implications on targeted and equitable air pollution mitigation plans in China.

## Results

**PM$_{2.5}$-related deaths attributable to household consumption**. Figure 1 shows the estimated shares of premature deaths due to anthropogenic PM$_{2.5}$ air pollution in China in 2012 according to consumption activities. Of the 1.08 (95% CI: 0.74–1.42) million deaths, 20% (212 thousand; 95% CI: 144–279) are related to direct emissions from fuel consumption in households for cooking and heating (hereinafter referred to as direct emissions), and 24% (271 thousand; 95% CI: 184–358) are related to emissions embodied in household consumption of goods and services (hereinafter referred to as indirect emissions). The remaining 56% of deaths are linked to emissions embodied in other consumption types (i.e., capital investment, government consumption, exports, and cross-boundary transport) and are not discussed further in this work. The majority of deaths related to direct emissions occur in rural household due to massive use of solid fuels (e.g., crop residues, wood, and coal) for cooking and heating in rural

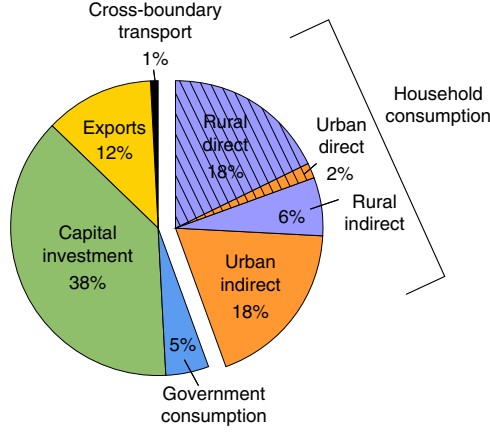

**Fig. 1** Outdoor PM$_{2.5}$-related premature deaths attributed to sources and consumption activities in 2012 in China. Total number of PM$_{2.5}$-related premature deaths attributed to anthropogenic emissions, and their attribution to final demand categories from supply chain perspective. Source data are provided as a Source Data file

areas, while deaths related to indirect emissions are dominated by urban household consumption. On average, the number of deaths attributed to rural consumption is higher than that related to urban consumption when considering both direct and indirect emissions.

**Household consumption related deaths of 12 income groups**. Figure 2 and Supplementary Table 1 show the relationship of income and air pollution-related deaths due to consumption by settings (i.e., rural or urban) and emission types (i.e., direct or indirect). For each setting, bars in Fig. 2 are ordered from the poorest on the left to the richest on the right (per capita income plotted in gray line above in Fig. 2). For direct emissions, poor rural residents are related to more deaths than richer rural residents as poorer households tend to consume more solid fuel than richer households. Deaths due to direct emissions of extremely poor rural residents are 20% greater than those due to direct emissions of high-income rural residents (Fig. 2), and the poorest 10% of rural residents are related to 21% of air pollution-related deaths from direct emissions (see Fig. 3c). Compared with rural households, direct emissions from urban households have much less health impacts and are less correlated with income levels because most Chinese urban households have access to clean fuels[32]. Health impacts associated with indirect emissions increase remarkably with income level for both rural and urban households. Residents in the highest income brackets of rural and urban areas are related to 2.3 and 3.5 times more deaths than residents in the lowest income groups, respectively. This phenomenon has also been reported on other environmental footprints from household consumption, such as carbon emissions[25] and water use[19], which could be attributed to resource-intensive consumption patterns of rich families.

Taking direct and indirect emissions both into account, residents in the highest income brackets of rural and urban areas are related to 1.1 and 3.3 times more deaths than residents in the lowest income groups, respectively. Figure 3 further shows the inequality of air pollution-related deaths due to consumption and income earned by Chinese households in 2012. The richest 10% of residents earn 29% of total household income and are linked to 13% of air pollution-related deaths. In comparison, the poorest 10% of residents earn only 1% of total household income yet are linked to 11% of air pollution-related deaths. While the

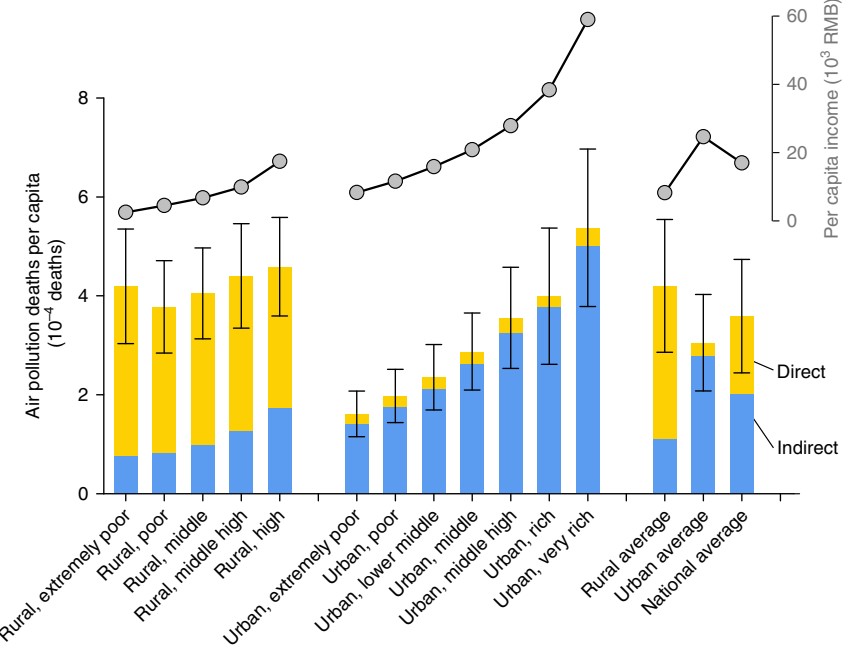

**Fig. 2** Per capita intensity for household consumption related premature deaths of 12 income groups. The income groups are ordered from the poorest on the left to the richest on the right for rural and urban households, respectively. The per capita income for each income group are shown in gray dot lines. Error bars present uncertainty ranges (95% CI) of the estimates. Source data are provided as a Source Data file

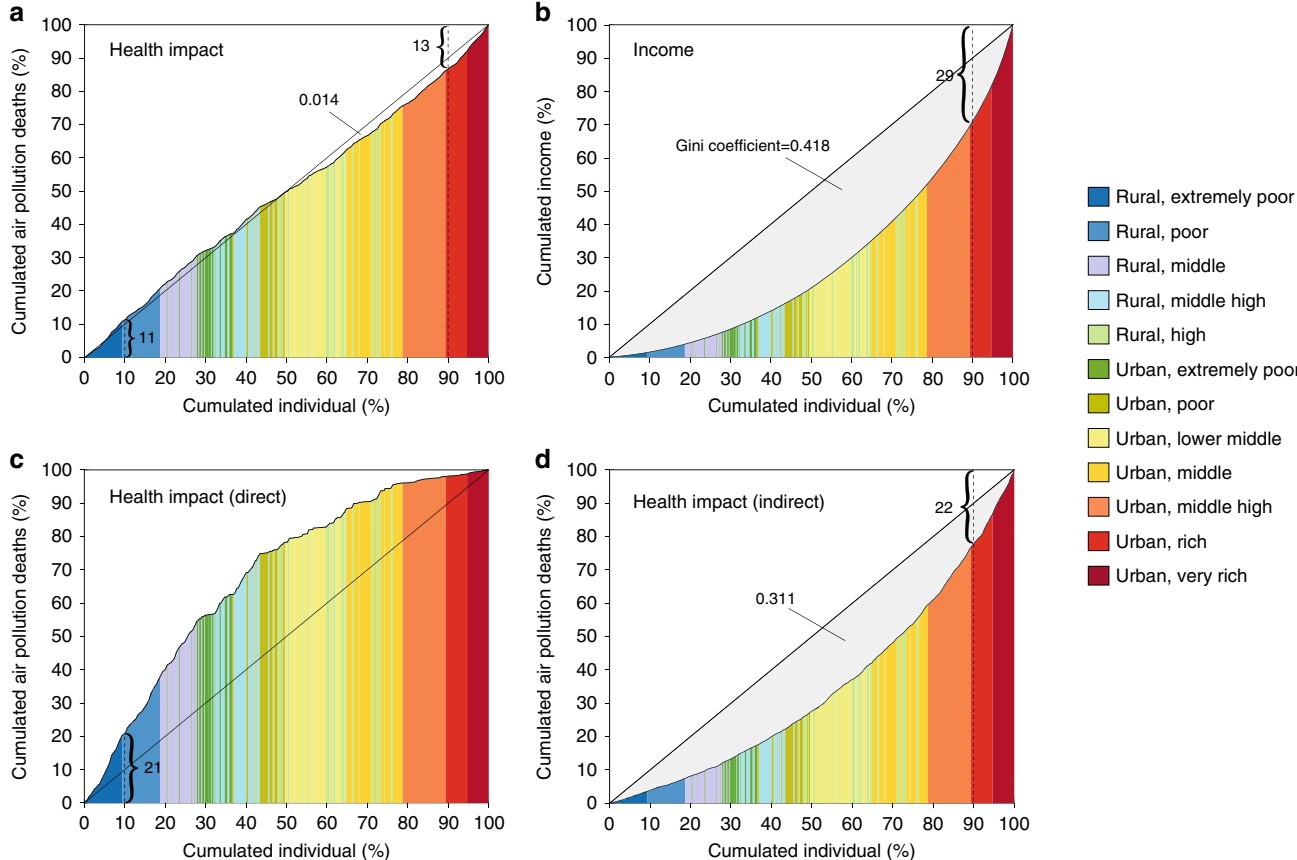

**Fig. 3** Lorenz curves for air pollution-related deaths attributable to consumption and income earned for households. **a** Lorenz curve for premature deaths related to household consumption emissions. **b** Lorenz curve for household income earned. **c** Lorenz curve for premature deaths related to household direct emissions. **c** Lorenz curve for premature deaths related to household indirect emissions. Household groups are sorted by income per capita ordered from the poorest on the left to the richest on the right, and the horizontal axis in panel (**a**–**d**) show the cumulative population share. The diagonal is the line of perfect equality. Source data are provided as a Source Data file

distribution of indirect impacts is consistent with the income (Fig. 3c), the impacts from direct emissions are inversely distributed to income level (Fig. 3d), meaning that air pollution-related deaths are more evenly distributed than income earned nationwide; whereas the Gini coefficient in 2012 was 0.418 (Fig. 3b), the analogous inequality coefficient of air pollution-related deaths in the same year is only 0.014 (Fig. 3a). Differences in consumption patterns also affect the spatial distribution of air pollution-related deaths.

**Regional differences in household consumption related deaths.** Figure 4 shows the differences in air pollution-related deaths attributed to household consumption in seven different regions of China (see region definitions in Supplementary Table 2 and Supplementary Fig. 1). Regions are ordered from the poorest average per capita income on the top to the richest at the bottom (Fig. 4a). Air pollution-related deaths due to household emissions in each region are shown by the bars in Fig. 4c, with local (within region) deaths indicated on the left, and the magnitude and location of deaths resulting in other regions indicated by the colored bars on the right. The disparity between numbers of regional deaths from direct emissions could be explained by variety in meteorological conditions[33] and differences in solid fuel consumption due to different temperature and income levels[34,35]. Health impacts from direct emissions are higher than that from indirect emissions in poorer and colder regions such as Southwest, Northwest, and Northeast, while impacts from direct emissions are significantly lower than that from indirect emissions in east coastal regions (Yangtze River Delta and Southeast) which are richer and warmer. Cross-regional impacts from direct emissions are more concentrated in downwind regions. In contrast, indirect emissions lead to more broad cross-regional health impacts than direct emissions due to widespread supply chains across the whole country: 48% of deaths related to household indirect emissions occur in a different region from where the household consumption occur, with this share ranging from 31% in the Southwest to 65% in the Northwest (Fig. 4c). Among different regions, consumption of goods and services in Central, North, and Yangtze River Delta regions are linked to the most deaths due to massive consumption in these regions (Supplementary Fig. 2). Consumption in the relatively poor Northwest

region nonetheless results in substantial deaths in other regions because its indirect emissions are concentrated in population-dense areas such as the North and Central regions.

**Discussion**

This work develops the quantitative relationship between household consumption and air pollution-related premature deaths for the first time. Using the newly established method, we separate the air pollution-related deaths from direct and indirect consumption for urban and rural residents in China. Although substantial contribution of solid fuel use on air pollution in China has been investigated[36–38], we find unexpected higher contribution of rural household consumption to air pollution-related deaths in China. These findings further emphasize the great importance of mitigating emissions from direct emissions of rural households, given that current policies focus more on urban pollution. Indeed, our results likely underestimate air pollution-related deaths by rural households because of neglecting the impacts of indoor air pollution from solid fuel use. Policies that promote clean energy (e.g., natural gas and electricity use) in rural households could provide a perfect solution, however, the high prices, lack of accessibilities to natural gas, and traditional consumption behaver might hinder the promotion of such policy[39]. Because urban residents also suffer the pollution from rural emissions and have higher willingness to pay for alleviating pollution, providing price subsidy within a certain time period might be a possible solution given that urban residents pay more taxes than rural residents. The price of clean energy will be eventually accepted by rural residents with economy developed and income increased.

Our results indicate that income and thus the scale of household expenditures are closely related to the air pollution-related deaths related to the consumption of urban households in China. For example, we estimate that 10,000 very rich urban consumers account for 5.4 premature air pollution-related deaths (95% CI: 3.81–7.0) per year—a factor of 3.3 times more than 10,000 extremely poor urban consumers (1.6 deaths, 95% CI: 1.2–2.1). This is within the line of other studies related to income and environmental impacts; impacts of rich urban consumers are higher than poor consumers by a factor of 3.8–9.5 for different environmental indicators (i.e., $CO_2$ emissions, air and water

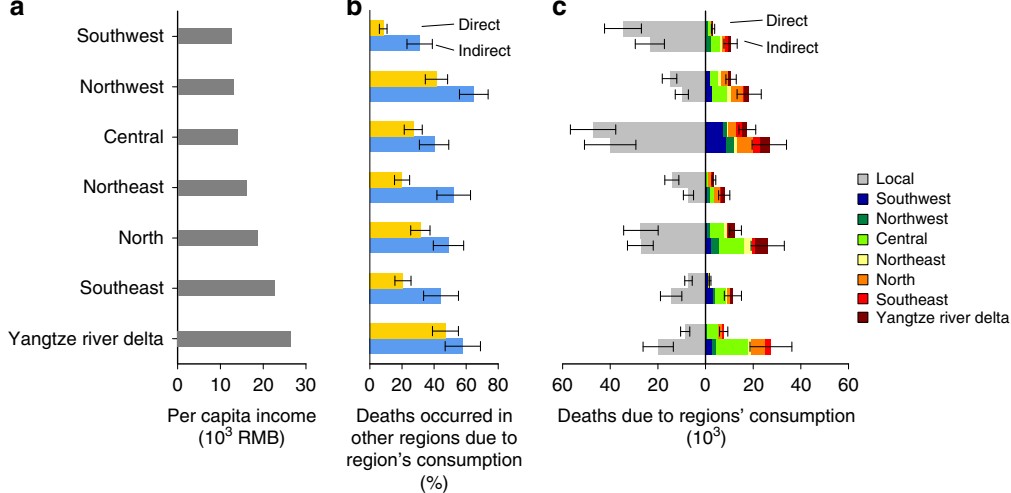

**Fig. 4** Premature deaths related to consumption in seven regions of China. **a** Per capita income of seven regions. **b** Ratio of deaths occurred in other regions due to each region's direct (yellow) and indirect (blue) consumption. **c** Regional household consumption related deaths and where the deaths occurred. For panel (**c**), the gray bar extend to the left means deaths occurred locally and the colored bar extend to the right means the deaths occurred in other regions. Error bars in panels (**b**) and (**c**) present uncertainty ranges (95% CI) of the estimates. Source data are provided as a Source Data file

pollutant emissions, and water use)[19,24,25,40]. Our work provides additional insight to this discussion by adding air pollution-related premature deaths as a new indicator.

Our work provides unprecedented quantitative insight into the supply chain patterns that link final consumption to air pollution-related premature deaths. For example, by tracking the health impacts along the supply chains, we show that the cross-regional health impacts of emissions embodied in household consumption of goods and services are much greater than the effects of atmospheric transport across regional boundaries. Moreover, we highlight systematic differences in the impacts of household consumption on residents according to their income level and location. These findings point to targeted opportunities for pollution abatement, such as direct emissions from solid fuels burned by rural households, but more importantly offer a basis for clean air policies that avoid and redress socio-economic and regional inequities. Reducing emissions throughout the supply chains will require some combination of improved air pollution control technologies, changes in energy mix, and changes in the location of manufacturing[41]. To the extent these changes must be undertaken by less economically developed regions and households, consumption-based policies may better support the needed technology transfer and capital investment while at the same time encourage more sustainable consumption behaviors.

Our study is subject to a number of uncertainties and limitations from the use of multiple datasets and complex models. A detailed, quantitative uncertainty analysis for each step of this study is conducted and presented in Supplementary Discussion and the overall uncertainty ranges (95% CI) associated with mortality estimates are presented in Figs. 2 and 4. First, bottom-up emission inventories are uncertain due to the lack of complete data of activity rates and local-measured emission factors[42]. The MEIC emission inventory used in this study has been widely applied in chemical transport models and validated against observations[43,44]. Second, incomplete income and expenditure data at provincial level contribute to the uncertainties in estimating emissions consumed by each income group. Improvement of statistics reporting system or conducting filed surveys could remedy this situation in the future. Third, sensitivities of emissions to PM$_{2.5}$ exposures are simulated from the GEOS-Chem model and its adjoint, which are also subject to uncertainties due to incomplete knowledge of chemical and physical processes. We compare the modeled and the satellite-derived PM$_{2.5}$ concentrations and reasonable correlations are found for most regions in China (Supplementary Fig. 3). Last but not least, the IER function used in mortality estimates are developed based on cohort studies in western countries and may introduce additional uncertainties when applying for China due to differences in PM$_{2.5}$ toxicities, population adaption, and healthcare levels. Using concentration–response relationships developed from local cohort studies could improve the estimates of premature mortalities in the future.

## Methods

**Integrated modeling framework**. This work combines data and models from multiple sources to quantify the premature deaths due to PM$_{2.5}$ pollution related to household consumption activities of 12 income groups in 30 provinces of mainland China, as depicted in Supplementary Fig. 4. The datasets used in this study include a bottom-up emission inventories of major air pollutants obtained from the Multi-resolution Emission Inventory of China (MEIC[39,45]: http://www.meicmodel.org/), the Multi-Regional Input–Output (MRIO) table of China from Mi et al.[46], income and expenditure data over China at national and provincial level from national and provincial statistical yearbooks[47–49], as well as satellite-based ground-level PM$_{2.5}$ mass concentrations over China from Geng et al.[50]. All the data are for the year of 2012.

**Production-based emission inventory**. The MEIC model, which is developed and maintained by Tsinghua University, provides the production-based anthropogenic

emissions of sulfur dioxide (SO$_2$), nitrogen oxides (NO$_x$), ammonia (NH$_3$), black carbon (BC), organic carbon (OC), and anthropogenic PM$_{2.5}$ dust used in this study. These atmospheric pollutant emission inventories are used to estimate the direct and indirect emissions from Chinese household consumption, and as the inputs for GEOS-Chem model and its adjoint.

The MEIC is a bottom-up emission inventory framework which covers 31 provinces in mainland China and includes more than 700 anthropogenic emitting sources. It is improved based on the bottom-up emission inventory developed by the same group[42], which uses technology and process-based methods to resolve the quantitative relationship between emissions and technology turnover. Detailed description of the technology-based methodology and the source classifications can be found elsewhere[39,45]. The sources of the underlying data used in the MEIC model, including the activity rates, technology penetration data, and emission factors, are summarized in Zheng et al.[39].

**Direct and indirect emissions from household consumption**. Emissions caused by household consumption activities come from both household direct energy use (fuel combustion for home cooking and/or independent heating, and private car; i.e., direct emissions) and their expenditure on goods and services which use energy and other resources as intermediate inputs (i.e., indirect emissions)[51].

Household direct emissions by region or provinces can be obtained from the MEIC model directly. For rural households, direct emissions include residential biomass/fossil fuel combustion, and private car emissions; for urban households, direct emissions only include fossil fuel combustion, and private car emissions. All direct emissions are emitted locally.

Household indirect emissions are produced and emitted throughout the supply chains among sectors and regions who take part in the production process of household consumed goods or services. Here, we use the MRIO model of China[46] to attribute provincial emissions to household consumption. The MRIO table includes 30 provincial-level administrative divisions (Tibet, Macao, Hong Kong, and Taiwan are not included) and 30 aggregated sectors. Detailed information about the 30 sectors are provided in Supplementary Table 3.

The MRIO analysis starts with the monetary flows between sectors and regions:

$$\begin{pmatrix} \mathbf{x}^1 \\ \mathbf{x}^2 \\ \mathbf{x}^3 \\ \vdots \\ \mathbf{x}^{30} \end{pmatrix} = \begin{pmatrix} \mathbf{A}^{1,1} & \mathbf{A}^{1,2} & \mathbf{A}^{1,3} & \cdots & \mathbf{A}^{1,30} \\ \mathbf{A}^{2,1} & \mathbf{A}^{2,2} & \mathbf{A}^{2,3} & \cdots & \mathbf{A}^{2,30} \\ \mathbf{A}^{3,1} & \mathbf{A}^{3,2} & \mathbf{A}^{3,3} & \cdots & \mathbf{A}^{3,30} \\ \vdots & \vdots & \vdots & \ddots & \vdots \\ \mathbf{A}^{30,1} & \mathbf{A}^{30,2} & \mathbf{A}^{30,3} & \cdots & \mathbf{A}^{30,30} \end{pmatrix} \begin{pmatrix} \mathbf{x}^1 \\ \mathbf{x}^2 \\ \mathbf{x}^3 \\ \vdots \\ \mathbf{x}^{30} \end{pmatrix} + \begin{pmatrix} \sum_s \sum_t \mathbf{y}_t^{1,s} \\ \sum_s \sum_t \mathbf{y}_t^{2,s} \\ \sum_s \sum_t \mathbf{y}_t^{3,s} \\ \vdots \\ \sum_s \sum_t \mathbf{y}_t^{30,s} \end{pmatrix} \quad (1)$$

where $\mathbf{x}^r$ is the vector of total economic output for each sector in province $r$; $\mathbf{A}^{r,s}$ is the direct requirement coefficient matrix in which the columns reflect the input requirement by sector in region $r$ to produce one unit of output of the sector in region $s$; $\mathbf{y}_t^{r,s}$ is the final demand vector of category $t$ for each sector that are finally produced in region $r$ and consumed in region $s$. Here $t = 1, 2\cdots5$, means rural household consumption, urban household consumption, government consumption, capital investment, and exports, respectively. Equation (1) can also be abbreviated as:

$$\mathbf{x} = \mathbf{A}\mathbf{x} + \mathbf{y} \quad (2)$$

where $\mathbf{x}$, $\mathbf{A}$, and $\mathbf{y}$ are the block matrix or vector in Eq. (1). Solving for total output we can get:

$$\mathbf{x} = (\mathbf{I} - \mathbf{A})^{-1}\mathbf{y} \quad (3)$$

where $\mathbf{I}$ is the identity matrix, and $(\mathbf{I} - \mathbf{A})^{-1}$ is the Leontief inverse matrix.

Combined with the emission intensity by sector, pollutant emissions embodied in the trade flow can be calculated as:

$$\mathbf{e} = \widehat{\mathbf{f}}(\mathbf{I} - \mathbf{A})^{-1}\mathbf{y} \quad (4)$$

where $\widehat{\mathbf{f}}$ is the diagonalization of the vector of region-specific pollutant emissions for unit output by sector. The region-specific pollutant emissions used to produce $\widehat{\mathbf{f}}$ are obtained from the MEIC model, and the mapping process between sectors defined in the MEIC inventory and the MRIO model for each province can be found in our previous studies[15,17].

Then, region- and sector-specific emissions attributed to final demand $t$ in region $s$ can be calculated as:

$$\mathbf{e}_t^s = \widehat{\mathbf{f}}(\mathbf{I} - \mathbf{A})^{-1} \begin{pmatrix} \mathbf{y}_t^{1,s} \\ \vdots \\ \mathbf{y}_t^{r,s} \\ \vdots \\ \mathbf{y}_t^{30,s} \end{pmatrix} \quad (5)$$

where $\mathbf{e}_t^s = (\mathbf{e}_t^{1,s}, \mathbf{e}_t^{2,s}, \mathbf{e}_t^{3,s} \dots \mathbf{e}_t^{30,s})$; $\mathbf{e}_t^{r,s}$ is a sector-specific vector for emissions occurred in region $r$ caused by final demand $t$ in region $s$; $\mathbf{y}_t^{r,s}$ is the finished products produced in region $r$ consumed in region $s$ belonged to category $t$.

Then, total emissions from household consumption can be written as:

$$\mathrm{ce}_t^s = \sum_r \sum_l e_{l,t}^{r,s} + \mathrm{de}_t^s \qquad (6)$$

where $\mathrm{ce}_t^s$ is the total emissions from household consumption of region $s$ for final demand $t$ (rural or urban household consumption); $e_{l,t}^{r,s}$ is the emissions of sector $l$ in region $r$ caused by final demand $t$ in region $s$, and $\mathrm{de}_t^s$ is the household direct emissions in region $s$ for final demand $t$.

**Tracing household consumption emissions to income groups.** In this section, we trace the estimated emissions from urban and rural household consumption to various income groups according to their expenditure on daily consuming products or direct energy consumption.

The income and expenditure data used in this study are obtained from national and provincial statistic yearbooks[47–49]. The statistical yearbooks report average incomes and consumption expenditure patterns for different income groups in a province based on the sampling survey conducted by the National Bureau of Statistics of China. Usually, the total households are split into 12 income groups (7 urban and 5 rural) according to the household numbers. The seven urban income groups are extremely poor (10% of the urban household number), poor (10%), lower middle (20%), middle (20%), middle high (20%), rich (10%), and very rich (10%). The five rural income groups are poor (20% of the rural household number), lower middle (20%), middle (20%), middle high (20%), and rich (20%).

The income and expenditure data from statistic yearbooks have two limitations that need to be adjusted before our analyses. First, these data are not available for all the 30 provinces considered in our study. In 2012, only 90% and 43% provinces report the average incomes by groups for urban and rural households, respectively; only 83% and 40% provinces report the consumption expenditure patterns by groups for urban and rural households, respectively. Second, there are inconsistency in the average incomes of the income groups between different provinces, because regions or provinces in China experience different development stages. For example, the poor group of rural household in Beijing has similar average income value with the middle group of rural household in Heilongjiang. This might introduce biases when conducting a national analysis.

To solve the problems mentioned above, first, we make an assumption that those provinces with no grouped income or expenditure data have similar income or expenditure patterns with the national average or their neighboring provinces. Using the province-averaged income data adopted from the national statistical yearbook as scale factor, and the grouped income data from the national average or the neighboring provinces as proxies, we split the ungrouped data into 12 groups. Results are shown in Supplementary Fig. 5. Similar approach are used for the consumption expenditure data. Second, we range all income groups from 30 provinces according to their average incomes for urban and rural separately. Following the protocol by the statistic yearbooks, we allocate them into seven urban and five rural groups according to the resident numbers (Supplementary Fig. 6) instead of the household numbers. The resident numbers of each income groups are calculated using the following equation:

$$\mathrm{rn}_s^j = \frac{P_s}{\mathrm{pph}_s} \times \mathrm{frac}_s^j \times \mathrm{pph}_s^j \qquad (7)$$

where $\mathrm{rn}_s^j$ is the resident number of income group $j$ in province $s$; $P_s$ is the total urban or rural population in province $s$; $\mathrm{pph}_s$ is the province-average persons per household for province $s$; $\mathrm{pph}_s^j$ is the average persons per household of income group $j$ in province $s$; $\mathrm{frac}_s^j$ is the household fraction (10 or 20%) of income group $j$ in province $s$.

Using the consumption expenditure patterns for different income groups in each province, we can trace the indirect emissions estimated in the previous section to different income groups. The national statistical yearbook has relatively detailed sectors for household consumption, while provincial statistical yearbooks only report limited aggregated sectors. Therefore, we first allocate the aggregated sectors from the provincial data into detailed sectors using national data as proxies. Then we map them into the 30 sectors in the MRIO table. The mapping processes can be found in Supplementary Table 3, which is similar to that in Wiedenhofer et al.[25]. Sector-specific per capita household expenditure of various income groups by provinces for rural and urban, separately:

$$\mathbf{v} = \begin{bmatrix} \mathbf{v}_1^1 & \mathbf{v}_1^2 \\ \mathbf{v}_2^1 & \mathbf{v}_2^2 \\ \vdots & \vdots \\ \mathbf{v}_{30}^1 & \mathbf{v}_{30}^2 \end{bmatrix} \qquad (8)$$

where $\mathbf{v}_s^1$ is a $30 \times 5$ matrix for sector-specific per capita expenditure of five rural income groups in province $s$; $\mathbf{v}_s^2$ is a $30 \times 7$ matrix for sector-specific per capita expenditure of seven urban income groups in province $s$. All these data are based on the original groups of each province.

Rural and urban household consumption of region $s$ can be split into income group $i$ as:

$$\mathbf{y}_{1,i}^{r,s} = \left[ \mathbf{n}_s^i (\mathbf{v}_s^1)' / \sum_{i=1}^5 (\mathbf{n}_s^i (\mathbf{v}_s^1)') \right]' * \mathbf{y}_1^{r,s}, \quad i = 1 \cdots 5 \qquad (9)$$

$$\mathbf{y}_{2,i}^{r,s} = \left[ \mathbf{u}_s^i (\mathbf{v}_s^2)' / \sum_{i=1}^7 (\mathbf{u}_s^i (\mathbf{v}_s^2)') \right]' * \mathbf{y}_2^{r,s}, \quad i = 1 \cdots 7 \qquad (10)$$

where $\mathbf{y}_{1,i}^{r,s}$ is the sector-specific finished products produced in region $r$ and consumed by rural households in region $s$ which are allocated to the new national income group $i$. $\mathbf{n}_s^i = [n_s^{1,i} \; n_s^{2,i} \; n_s^{3,i} \; n_s^{4,i} \; n_s^{5,i}]$ is the rural population in region $s$ assigned to the new national group $i$, and its element $n_s^{j,i}$ means people belonged to the original group $j$ in region $s$ and are allocated to national income group $i$. Similarly, $\mathbf{u}_s^i$ and $u_s^{j,i}$ have the counterpart meanings for urban households. The $*$ means the hadamard product of the two vectors. Emission embodied in the corresponding groups can be calculated as:

$$\mathbf{e}_{1,i}^s = \widehat{\mathbf{f}}(\mathbf{I} - \mathbf{A})^{-1} \begin{pmatrix} \mathbf{y}_{1,i}^{1,s} \\ \vdots \\ \mathbf{y}_{1,i}^{r,s} \\ \vdots \\ \mathbf{y}_{1,i}^{30,s} \end{pmatrix}, \quad i = 1 \cdots 5 \qquad (11)$$

$$\mathbf{e}_{2,i}^s = \widehat{\mathbf{f}}(\mathbf{I} - \mathbf{A})^{-1} \begin{pmatrix} \mathbf{y}_{2,i}^{1,s} \\ \vdots \\ \mathbf{y}_{2,i}^{r,s} \\ \vdots \\ \mathbf{y}_{2,i}^{30,s} \end{pmatrix}, \quad i = 1 \cdots 7 \qquad (12)$$

where $\mathbf{e}_{t,i}^s = (\mathbf{e}_{t,i}^{1,s}, \mathbf{e}_{t,i}^{2,s}, \mathbf{e}_{t,i}^{3,s} \dots \mathbf{e}_{t,i}^{30,s})$, and $\mathbf{e}_{t,i}^{r,s}$ is a sector-specific vector for emissions occurred in region $r$ caused by rural ($t = 1$) and urban ($t = 2$) households in region $s$ which belongs to national income group $i$.

For household direct emissions, different approaches are used for different emission sources when tracing them into income groups. For urban household fossil fuel and private car emissions, we use the household expenditure on residential energy consumption (residence for rural household listed in Supplementary Table 3) and the emissions from urban and rural household consumption (transport and communication in rural household), respectively, as proxies to allocate them into various income groups, similar as the process for indirect emissions (Eqs. (9–12)). For rural biomass consumption emissions, we use a correlation equation between biomass consumption and income per capita adopted from Peng et al.[34] to allocate the emissions into various rural income groups. This correlation equation is fitted using hierarchical regression based on the survey-based per capita income and biofuel consumption, which is shown below:

$$t_{\mathrm{bio},j} = 0.7072 \times \alpha_j^{-0.18} \qquad (13)$$

where $\alpha_j$ is the per capita income of income group $j$; $t_{\mathrm{bio},j}$ is the biomass consumption for people at the income group $j$. Usually, the poor households tend to consume more biomass and less commercial fuel[52]. More details about this correlation equation can be found in Peng et al.[34].

Then rural household biomass combustion emissions in region $s$ can be allocated to national group $i$ as:

$$e_{\mathrm{bio},i}^s = \mathbf{n}_s^i (\mathbf{t}_{\mathrm{bio}}^s)' / \sum_{i=1}^5 (\mathbf{n}_s^i (\mathbf{t}_{\mathrm{bio}}^s)') * e_{\mathrm{bio}}^s, \quad i = 1 \cdots 5 \qquad (14)$$

where $\mathbf{t}_{\mathrm{bio}}^s = [t_{\mathrm{bio},1}^s \; t_{\mathrm{bio},2}^s \; t_{\mathrm{bio},3}^s \; t_{\mathrm{bio},4}^s \; t_{\mathrm{bio},5}^s]$ is the per capita biomass consumption of different income groups. Note that $t_{\mathrm{bio},j}^s$ used here is based on per capita income of the original groups in each province. $e_{\mathrm{bio}}^s$ indicates the biomass consumption related emissions of rural households in region $s$.

Based on the allocating processes described above, we finally get the province- and sector-specific emissions induced by household consumption of income group $i$ in region $s$: $\mathbf{ce}_{t,i}^s$. Then, emissions attributed to income group $i$ in region $s$ can be allocated to grid cells based on the sector-specific spatial distribution from the MEIC inventory. The attributed ratios are:

$$\beta_{t,i,k}^{r,s} = \mathbf{ce}_{t,i,k}^{r,s} . / \mathbf{e}_k^r \qquad (15)$$

where $\mathbf{e}_k^r$ is the sector-specific emission vector for species $k$ produced in region $r$; $\boldsymbol{\beta}_{t,i,k}^{r,s}$ is the sector-specific ratios of emissions occurred in region $r$ induced by household consumption of income group $i$ in region $s$.

As mentioned above, income group and expenditure pattern data are missing for some provinces in the statistical yearbooks, and we use data from the national average or the neighboring provinces to estimate the grouped data in these provinces. This

assumption might introduce uncertainties in our results. To determine the uncertainties brought by such assumption, we conduct several sensitivity scenarios. In each scenario, we use data from the national average or one of the provinces that have available statistical data to estimate the grouped data for all the provinces with missing data. Urban and rural cases are treated separately. In total, we have 26 scenarios for urban household and 13 scenarios for rural household, as 25 and 12 provinces report income data by group for urban and rural households, respectively. We use the estimated income group data to calculate direct and indirect emissions related to each income group. The coefficient of variation (CV) of the estimated direct and indirect emissions in each income group are shown in Supplementary Table 4. In general, the CV values are within ±10%, which means the uncertainties introduced by the assumptions in our study are limited and our method is robust.

**Estimating PM$_{2.5}$-related premature deaths**. In this study, we use satellite-based ground-level PM$_{2.5}$ mass concentrations and the IER model from GBD 2010 to estimate PM$_{2.5}$-related premature deaths.

Satellite-derived PM$_{2.5}$ concentrations provide relatively accurate scale and spatial distribution for PM$_{2.5}$ exposure[53]. The satellite-based PM$_{2.5}$ concentration data used in this study are obtained from our previous study[50], which are estimated using the aerosol optical depth (AOD) derived from satellite instruments (MODIS and MISR onboard the Terra satellite) and conversion factors between AOD and PM$_{2.5}$ simulated by the GEOS-Chem chemical transport model[54].

The IER model is developed by Burnett et al.[30], and has been used to estimate the PM$_{2.5}$-related premature deaths in previous studies[4,5,31]. In Cohen et al.[55], an updated version of the function are provided, yielding about 35% higher mortality estimates compared with previous works. Therefore, the results in our study present the lower limits of the estimates. The IER model describes the concentration–response relationship for the entire range of PM$_{2.5}$ concentration observed in the world, by incorporating data from cohort studies of ambient air pollution, first- and second-hand tobacco smoking, and household indoor air pollution[30]. Here we focus on the four leading causes of the PM$_{2.5}$-related premature mortality: ischemic heart disease (IHD), stroke, chronic obstructive pulmonary disease (COPD), and lung cancer (LC). For each disease, the relative risk (RR) is calculated as:

$$RR(C) = \begin{cases} 1 + \alpha\left(1 - e^{-\gamma(C-C_0)^\delta}\right), & \text{if } C > C_0 \\ 1, & \text{else} \end{cases} \quad (16)$$

where $C$ is the satellite-based annual mean PM$_{2.5}$ concentrations in 2012 at a $0.5° \times 0.667°$ resolution; $C_0$ is the counterfactual concentration and bellow which there is assumed to be no additional risk; $\alpha$, $\gamma$, and $\delta$ are parameters describing the overall shape of the concentration response. In this study, we use the parameters adopted from Lee et al.[31], and the values are listed in Supplementary Table 5. The mortality attributed to PM$_{2.5}$ pollution are estimated as:

$$M^{tot} = \frac{RR - 1}{RR} \times B \times P \quad (17)$$

where $M^{tot}$ is the total mortality related to PM$_{2.5}$; $\frac{RR-1}{RR}$ is the attributable fraction to PM$_{2.5}$ pollution; $B$ is the baseline incidence of a given health endpoint for all age group derived from the national average data in GBD 2013[56]; $P$ is the size of the exposed population aggregated from the LandScan global population database for 2012 at a 1 km resolution[57]. Value for $B$ used here can be found in Supplementary Table 5.

To determine the premature death attributable to anthropogenic sources, we use GEOS-Chem model to simulate the fraction of PM$_{2.5}$ pollutions contributed by anthropogenic emissions by conducting two scenarios: one with all emissions as inputs and the other turning off the anthropogenic emissions. Here, we use the direct proportion approach, which assumes a linear relationship between the proportions of total PM$_{2.5}$ concentration to the proportion of total mortality, to estimate the source-specific premature deaths. The scientific basis of this assumption has been validated by a GBD research, GBD MAPS[14]. Other studies also choose the direct proportion approach to solve the nonlinear problem[6,58,59]. Thus, the anthropogenic PM$_{2.5}$-related premature death can be calculated as:

$$M^{anth} = M^{tot} \times \frac{C_{all} - C_{no\_anth}}{C_{all}} \quad (18)$$

where $M^{anth}$ is the premature mortality related to PM$_{2.5}$ attributable to anthropogenic sources; $C_{all}$ is the annual mean PM$_{2.5}$ concentrations from the scenario with all emissions; $C_{no\_anth}$ is the annual mean PM$_{2.5}$ concentrations from the scenario with anthropogenic emissions turned off.

**Linking premature deaths to different income groups**. We then use the GEOS-Chem adjoint model to link air pollutant emissions in different income groups to premature deaths attributed to PM$_{2.5}$. The adjoint of GEOS-Chem is able to determine the response of PM$_{2.5}$-related mortality to changes in emissions of inorganic precursor gases (i.e., SO$_2$, NO$_x$, and NH$_3$), carbonaceous particles (i.e., BC and OC) and primary anthropogenic PM$_{2.5}$ dust[60,61]. It allows for efficient computation of the partial derivatives of a scalar model response with respect to input conditions (e.g., emission rates). Previous studies have used the adjoint of GEOS-Chem to quantify the response of PM$_{2.5}$ concentrations and air pollution mortality to emissions sources[31,62,63].

In this work, we use the nested version of GEOS-Chem adjoint over East Asia (11°S–55°N, 70°E–150°E) at a $0.5° \times 0.667°$ resolution, with boundary conditions from a global simulation at a $2° \times 2.5°$ resolution. Following Lee et al.[31], we define the cost function in the adjoint model as the anthropogenic PM$_{2.5}$-related premature deaths resulting from long-term exposure to PM$_{2.5}$[31], as calculated in Eq. (18). The outputs provided by the adjoint model are the partial derivatives of this cost function with respect to anthropogenic emissions in the simulation domain, which we refer to as the sensitivities of receptor region's PM$_{2.5}$-related premature deaths to emissions at all species, locations and times[31,62,64]. The species considered in our study are NH$_3$, SO$_2$, NO$_x$, BC, OC, and anthropogenic PM$_{2.5}$ dust, and the input anthropogenic emissions are adopted from the MEIC model.

Due to computational constraints of the GEOS-Chem adjoint simulations, we classify 30 provinces in China mainland into seven receptor regions (see Supplementary Table 2 and Supplementary Fig. 1) based on their economic development and climate zones, and a total of seven groups of simulations are conducted, one group for each receptor region. To further reduce the computation cost, the model is conducted for four months (January, April, July, and October, 1 month for each season) for each group, and the averaged results of the 4 months are used to represent the annual level in 2012.

Combined the sensitivity simulated by the GEOS-Chem adjoint model and the gridded emissions, we could obtain the semi-normalized sensitivity[32,33,36,37]:

$$SS_{m,n,k} = \frac{\partial M}{\partial E_{m,n,k}} \times E_{m,n,k} \quad (19)$$

where $m$, $n$, and $k$ are indices for longitude, latitude and species; $SS_{m,n,k}$ means the contribution of emissions for species $k$ at location $(m, n)$ to the total premature deaths of the receptor region[62,64]; $\frac{\partial M}{\partial E_{m,n,k}}$ is the sensitivity outputs from the adjoint model; $E_{m,n,k}$ is the emissions for species $k$ at location $(m, n)$.

We then normalize SS to calculate the percentage contribution of source-specific emissions to premature deaths:

$$P_{m,n,k} = \frac{SS_{m,n,k}}{\sum_m \sum_n \sum_k SS_{m,n,k}} \times 100\% \quad (20)$$

The normalization process isolates the contribution of gridded emission to receptor regions' PM$_{2.5}$-related premature deaths, which neglects the nonlinear response of PM$_{2.5}$ to emissions changes. This normalized marginal method has been used in previous studies to attribute global or national radiative forcing to sub-regions or species[38,39].

Results from the previous sections can be integrated to attribute regional- and source-specific PM$_{2.5}$ deaths to household consumption of income group $i$ in region $s$:

$$M_{t,i}^s = \sum_r M^r \sum_p \sum_k \left( P_{(m,n)\in p,k}^r \times \beta_{(m,n)\in p,t,i,k}^{p,s} \right) \quad (21)$$

where $M^r$ the total premature deaths occurred in region $r$; $\beta_{(m,n)\in p,t,i,k}^{p,s}$ is sector average ratios of emission occurred in grid $(m, n) \in p$ within the simulation domain and were attributed to rural (when $t = 1$) or urban (when $t = 2$) household consumption of income group $i$ in region $s$.

**Lorenz curve and Gini coefficient for inequality measurement of health impact**. Lorenz curve was developed by Lorenz in 1905 to represent the inequality of wealth distribution among population[65]. It is presented as cumulative share of income earned (%) on the vertical axis versus cumulative share of people from lowest to highest incomes (%) on the horizontal axis. In recent years, it has been widely used to measure inequality in areas of energy and climate change[66,67]. Here we utilize Lorenz curve to represent the inequality of household consumption related premature deaths, see Fig. 3 and Supplementary Fig. 7. In the context of health impact here, we ranked the rural and urban household groups of different income level in 30 provinces from lowest to highest per capita income groups, and then presented the cumulative share of people (%) on the horizontal axis versus their cumulative share of consumption relative heath impacts (%) on the vertical axis.

The Gini coefficient proposed by Gini is a numerically presentation of the inequality of income or wealth[65]. It is usually defined mathematically based on the Lorenz curve. The Gini coefficient $G$ is calculated as:

$$G = 1 - \left| \sum_{h=1}^N (H_{h+1} - H_h)(I_h + I_{h+1}) \right| \quad (22)$$

where $H$ is the cumulative share of population and $I$ is cumulative share of consumption related premature deaths. $H_h$ indicates the cumulated number of population in household groups from 1 to $h$ based on ranking list from lowest to highest per capita income and divided by the total population; $I_h$ indicates the corresponding cumulated consumption related premature deaths by household groups from 1 to $h$ and divided by the total premature deaths which were caused by national total household consumption.

## Data availability

The Multi-resolution Emission Inventory of China are available from http://www.meicmodel.org/; the Multi-Regional Input–Output model are available at https://doi.org/

10.6084/m9.figshare.c.4064285. The source data underlying Figs. 1–4 and Supplementary Figs. 2, 5–7 are provided as a Source Data file. The datasets generated during this study are available in the figshare repository with the identifier https://doi.org/10.6084/m9.figshare.9745337.

## Code availability

The code of GEOS-Chem adjoint model is available at http://adjoint.colorado.edu:8080/. The codes used for analyzing data are available from the corresponding author on reasonable request.

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

## Acknowledgements

This study was supported by the National Science Foundation of China (41625020 and 41629051), Postdoctoral Innovative Talent Support Program (BX20180164), and Chinese Postdoctoral Science Foundation (2019M650712).

## Author contributions

Q.Z. designed the research. H.Z., X.L., Y.L., and G.G. performed the research. L.P., M.L., B.Z., and H.H. processed emission data. L.Z. and D.K.H. developed the GEOS-Chem adjoint model. Z.M., Z.L., and D.G. provided the MRIO data. H.Z., G.G., S.J.D., Q.Z., and K.H. interpreted data. G.G., H.Z., S.J.D., and Q.Z. wrote the paper with input from all co-authors.

## Additional information

**Competing interests:** The authors declare no competing interests.

