## [Peer Review File · Nature Communications]

Reviewers' comments:

Reviewer #1 (Remarks to the Author):

Reviewer's report for

Inequality of household consumption and air pollution deaths in China

Suggested revision: major revision

General Impression

I have read this paper from the perspective of a reader who is not familiar with input-output analysis and the concept of footprinting/embodied impacts. For me there are two pressing issues with the current paper. Firstly, the results section appears unclear to me and I cannot follow what is actually presented, and secondly, I believe that the authors should provide more context for the results. Currently, there is a focus on the actual numerical results, but contextual information about how the numerical results can be interpreted (through examples or specific insights). More explanation will help the reader to understand the meaning of the results much better. Further, I think the paper is narrow in its exploitation of the topic and the results. Please see me detailed comments below.

Detailed feedback

Abstract

The key feature of this article is not made clear: This paper will provide unprecedented quantitative insight into the supply chain patterns that link final consumption to premature deaths. I think the fact that the authors are able to account premature deaths to consumption that is geographically removed from where the death occurs must be developed further. Further, the authors use the word *cause* to link final consumers to the premature deaths. I would recommend to use - throughout the article - different vocabulary for the actual cause of death (which is always local), and the underlying driver (for example final consumption of a good far removed from the place where the death occurs). For example, the *cause* of death could be air pollution, which is *accounted* to final consumption somewhere else. In my opinion, sentences such as *the richest 10% of Chinese 53 consumers cause 15% of all premature air pollution deaths* can easily be misinterpreted.

Line 58

This is in my opinion the correct use of the word *cause*, because air pollution is identified as the actual cause of the deaths.

Line 63

This is a key sentence in the article: direct impacts and embodied impacts are explained. I think the authors should stick to their introduced terms throughout the entire paper. I will use the words *direct* and *embodied* throughout this review. It may also be advisable to add one or two more sentences to clarify the concept of embodied air pollution for readers that are not familiar with it.

Line 70

Here the authors use the word *related*. I assume you mean *embodied*?

Line 75

Here the authors use the word *caused* for what I understand are in fact embodied pollutant emissions.

Line 81

From here onwards, I do get confused whether direct or embodied deaths are discussed. Are emissions also part of the remaining 11% of death causes? Please give some examples what the natural causes are. It would also be interesting to see how the deaths by natural causes per 1 million inhabitants compare to other nations or for example the OECD average. I assume that from here onwards all assessment is done for the 89% of deaths that are accounted to anthropogenic emissions, and the remaining 11% are not further discussed in his paper? Please clarify.

Line 85

Those 42% of the 89% of all deaths that are accounted to household consumption, is that embodied deaths, or are they caused by direct exposure such as fuel burning that you discussed earlier? The pie chart Fig 1 B lists the standard IO final demand categories, plus international transport. I assume you split the household consumption into rural and urban, which would make these 42% purely embodied. My understanding is that there is a strong proportion of non trade-related deaths through direct combustion of fuels. These cannot be included in this calculation of embodied deaths, correct? Please clarify, what you are discussing here.

Line 87

Here you explain that a proportion of those deaths are due to direct impacts. I assume this means that the intermediate transaction part of the satellite block of your MRIO model receives a certain percentage of those 89% of anthropogenically-driven deaths, and the remainder are assigned to final household consumption, correct? Then Fig 1 B must be a mix of direct and embodied deaths? Please clarify.

Fig 2

From a first view, it looks like the majority of deaths occur in very rich families. I do not understand the bars for each income class. Firstly, I would recommend putting all the rural statistics to the left of the chart, and urban statistics to the right and then sort both the urban and the rural bars according to income class. This might reveal trends that are currently not easy to spot. Secondly, each bar shows a *direct consumption* and *indirect consumption*, but the y-axis shows the *deaths per consumer* (I assume this is direct and embodied combined?). Does each bar show the directly suffered deaths, and the embodied deaths in each class that are suffered, or that are caused? The black dotted line shows the per capital income? This is not clear from the legend.

Line 101

The sentence *For example, deaths due to direct emissions of extremely poor rural consumers were 22% greater those caused by direct emissions of high-income rural consumers* should be linked to the statement you made before. At the moment, most of the statements in the Results section stand by themselves,

including this one.

Line 137

Here you are switching your focus towards the Lorenz Curve. How does that fit into the storyline of this section?

Fig 3

This entire figure needs to be revised. First of all, I would recommend that you motivate that you are taking a different angle in this part of the results section. You are now presenting spatial results. What is the y-dimension of this axis? Part A of the figure is clear (apart from the y axis). In part B it is unclear whether direct or embodied deaths are reported, and whether the deaths are driven or suffered. Part C's title *Deaths occurred in other regions* does not fit with the rest of the study. Are these embodied deaths in the final consumption of each respective region? The same points hold for Part D. If the y axis are in fact the region listed on the right hand of the graph, then it does not make sense to connect the dots with a line in parts A, C, and D.

Fig 4

What is the take away message from Figure 4? I can see that low income households suffer higher direct health impacts, but you already showed that before. I suggest you discuss how Part A, C, and D fit together, especially since Part D might appear surprising, because poorer households suffer less than richer ones in this curve. All in all I do question if Fig 4 offers any more insights.

Remainder of the Results section

The remainder of this section appears also as a compilation of data observations, but they lack a consistent storyline that connect the presented results. Also, as mentioned at the beginning of this review, the different statements should receive some contextual framing to highlight how the findings translate into the real world.

Discussions and Conclusions

In this section I would expect the key take-away message from this paper. The fact that income is related to impact has been shown countless times before, and does not qualify as a new insight, despite the topical nature of this study. Additionally, too many of the statements made in this section are speculative or lack evidence.

Sentence starting in line 158

Please rephrase and explain what you mean by *complicated*.

Sentence starting in line 168

I think this statement can also be derived without your study. You mentioned that a large number of deaths are caused by exposure to local fuel combustion. Reducing combustion of these fuels will therefore inevitably lead to reduction in deaths.

Line 177

I believe that you cannot discuss urbanisation in relation to the discussed health impacts without considering other factors such as transitioning to other fuels etc. This analysis as it stands is incomplete.

References

A lot of work has been done on sub-national MRIO and also pollution. However, the authors do not discuss their results in the light of previous works in great detail. I recommend that the authors undertake a more thorough literature review on the topic.

Reviewer #2 (Remarks to the Author):

Zhao et al.: Inequality of household consumption and air pollution deaths in China

General comment on Main text:

The main text is well-written and potentially interesting, however I found a number of unclear points for the analysis used in this study. I like the story line of the paper on how the pollution inequality has been occurred in China? On the other hand, the analysis was based on some important assumptions on the model analysis and data. I am wondering whether or not the discussion of the paper is well supported by the reliable quantitative analysis. For example, the proper sector correspondence between household expenditure survey data and MRIO data is crucial in estimating the sectoral and provincial PM_{2.5} emissions and health impacts. In addition, the household expenditure survey and income data should be well constructed in the Chinese provinces. Yes, this is the first attempt to do it and assumptions are necessary in doing it. The step-by-step analysis based on the reliable consumption statistics would be important in the valuable quantitative analysis. I would like to admire the authors in this ambitious study, however I am negative in publishing this paper in the Nature Communications. A more detailed household consumption and income survey data consistent with the MRIO data is crucial in analyzing the income and pollution inequalities.

Specific comment on Supplementary Information:

(1) Page 3, Line 69: Geng et al. (2015) provided the result of spatial distribution of the satellite-derived PM_{2.5} concentrations in China for 2006–2012. What result did you use in this study? What year? The inventory database is available to the public?

(2) Page 3, Line 73: The IER model was used to convert the PM_{2.5} concentrations into PM_{2.5}-related premature deaths. What are the values of the parameters, alpha, gamma, and delta? Please clarify them.

(3) Page 3, Line 89: Why you did not use the population database for 2012 that is consistent with the data of the annual mean PM_{2.5} concentrations in 2012?

(4) Page 3, Line 93: How did you set the value of non-anthropogenic emissions, C_{no anth}? The anthropogenic emissions can be calculated by the equation, C-C_{no anth}. Then inserting the value of C-C_{no anth} into the right-hand side of equation (1), C provides the relative risk based on the anthropogenic emissions, RR_{anth}. Eq. (1) is a non-linear equation, whereas Eq. (3) is a linear equation. It seems to me that Eq. (3) is something wrong, and you can straightforwardly use the non-linear equation of (1). Please clarify it.

(5) Page 4, Line 108: What is E_{i, j, k}? M is the function of the E? I see that a combined equation from three equations, (1), (2), and (3) includes some inconsistencies in the mathematical model. A critical assumption may be included there. I don't know about it.

(6) Page 4, Line 134-Page 5, Line 135: The data on household direct emissions is available to the public? What is the MEIC model? The authors say that 'these emissions attend to occurred locally.' What is the meaning of this sentence? Why locally? The emissions can be widely spread by the wind, therefore the atmospheric transport model can work.

(7) Page 5, Eq. (6): t can be subscript.

(8) Page 5, Lines 155-158: e is important. What is the database of e? Is it available to the public? What is the mapping process? Equation (9) provides the region-specific pollutant emissions embedded in final demand of China and exports.

(9) Page 5, Line 160: Equation (10) provides the region-specific pollutant emissions embedded in final demand category t of region s. Specifically, Eq. (10) captures that the household consumption of Beijing can induce pollutant emissions in Jilin. This emission modeling technique is NOT new and many studies used this footprint analysis. The pollutant emissions from Jilin embedded in Beijing can be used as the database of estimating PM_{2.5} related premature death by inserting the embedded PM_{2.5} emissions into the right-hand side of Eq. (1)? I am wondering how the footprint analysis was combined with the risk impact analysis. There is no explanation about the mathematical connection. The novelty is unclear!

(10) Page 6, Line 166: What is 'sum'? Sigma_s would be correct.

- (11) Page 6, Line 180: What is the sampling survey? Did you do it in this study? Please provide the detailed information on it.
- (12) Page 6, Lines 183-190: I cannot understand these sentences. What are the ungrouped provinces? Why did you use their average income and group data from their neighboring provinces? Is this assumption plausible? The detailed explanation would be important to readers. How the 'average' resident number of each income group was estimated? A simple calculus about it would be useful for readers. This is also unclear.
- (13) Page 6, Lines 195-197: What is the difference between origin group j and group i based on national grouping. Why this distinction is important in this study?
- (14) Page 6, Lines 198-202: The provincial per capita household expenditure on various goods or services was used in this study. Some provinces have no statistics about it. Therefore the data of their neighboring provinces or national average was used as a proxy data. There is not the detailed explanation. This assumption is strong. Table S3 shows the sector mapping between MRIO and household consumption statistics. How did you justify the corresponding table? To me, the corresponding table of Table S3 is very rough without any justifications and sector definitions. For example, consumers hardly buy metals smelting and pressing as durable goods directly. Why residence sector is corresponded to mining sector? Processed foods is not be classified as Agriculture, because they need more electricity than agricultural products such as rice etc. There are unclear sector correspondences.
- (15) Page 7, Lines 216-217: The two equations captures that household consumptions of income class i of province s affected PM_{2.5} emissions of other provinces indirectly. Again, how the emissions data were converted into the PM_{2.5}-related premature deaths? There is no mathematical explanation. The subscript t is the final demand category that you have already defined above.
- (16) Page 7, L228: What is the correlation equation? What is the database on the statistical analysis of the relationship between biomass consumption and income per capita? This equation is important in determining the direct emissions from each household income class.
- (17) Page 8, L253: This is the connection between the footprint analysis and the health impact analysis. It seems to me that Eq. (23) is a 'linear' impact model. My understanding is that the non-linear impact model of Eq. (1) can be used to convert the PM_{2.5} emissions into the health impacts, nevertheless the authors did not use the non-linear impact model based on the emissions embedded in household consumptions of each income class of each province. The benchmark health risk coefficients were estimated by using the satellite-derived PM_{2.5} concentrations and then the health risk coefficients were 'proportionally' used to estimate the health impacts embedded in household consumptions of each income class of each province. The marginal health impacts are exactly same over the emitted PM_{2.5} emissions? If my understanding is correct, this model assumption is really correct? Smaller (larger) additional emissions may induce smaller (larger) additional health impacts.
- (18) Pages 8-9, S6: This analysis is potentially interesting, however I am wondering whether or not the estimated results are reliable and robust empirically.

Reviewer #3 (Remarks to the Author):

This study examined the inequality of household consumption and air pollution-related deaths in 30 provinces of China. It is interesting and the statistical methods used are appropriate. However, I have one major concern about the uncertainty of the results, for that the methodology of this paper involved multiple models and procedures. Thus, the uncertainty should be fully considered and discussed. However, this important information is lacked in the result and discussion sections.

Other minor comments:

- Title, "air pollution deaths" should be "air pollution-related deaths". The authors should revise this problem through the paper.
- Discussion and Conclusions, this paper reported many interesting findings. Policy implications of these results are needed, for that this information is important for the policymaker and

governments. And comparison with previous studies on the same topic is also needed.

-Line 191, please provide references after the statement "As concluded by previous studies".

-Methods, Line 206, the reason of choosing four months (January, April, July and October) of simulations for the total year needs to be well justified.

-Methods, Line 204, please provide the reason why Tibet, Hong Kong and Macao were not included in this paper.

-Supplementary materials, Table S4, provide the results of 95% confidence interval for the consumption related deaths and consumption related deaths per capita.

-Supplementary materials, Figure S1, revise "Ya ngtze" to "Yangtze".

Some language errors. The authors should check this problem through the manuscript.

-Line 93, "from poorest on the left to richest on the right" should be "from the poorest on the left to the richest on the right".

-Line 102, "greater those" should be "greater than those".

Reviewer #4 (Remarks to the Author):

This is interesting work with potentially important results. However, a central message is missing, and it is not entirely clear what part of the results agrees/disagrees with previous studies, and what part is novel and remarkable enough to justify a high-profile publication. Somehow, the article needs a "punch-line", and more pronounced indications of policy implications. I recommend that publication in Nature Comm. Is considered after major revision of the manuscript.

Major comments:

The calculations have been performed using IER functions of Burnett et al. (ref. 19), indicating that this is the latest model (supplement I.66). This is not correct, as the latest global burden of disease evaluation uses updates by Cohen et al. (2017), yielding about 35% higher mortality estimates compared to previous work. While I do not suggest repeating all calculations with the new IERs, it will be needed to discuss this and indicate that present estimates are lower limits.

An important result is that a similar number of deaths were caused by rural and urban consumption, and that residential energy use is a leading cause. I believe that this aspect could be highlighted more strongly, as discussions on air pollution typically focus on the urban environment and the role of traffic.

On the other hand, the result that the average income of those dying prematurely is often lower than that of the consumers causing their deaths is not remarkable.

The conclusion that "reducing solid fuel consumption by rural households and encouraging sustainable consumption in urban areas may be the most effective means of reducing the environmental and health inequalities related to Chinese air pollution" appears relevant. However, the part "environmental and health inequalities" could be replaced by "health consequences". On the other hand, is this really new? The last section writes "substantial reductions in air pollution deaths are possible by reducing the combustion of household solid fuel in China", which is not surprising.

The last paragraph (l.173-181) seems trivial.

Minor comments:

Although the text generally reads well, a few language mistakes (grammar) need to be corrected.

I.89: Please define "indirect" emissions more clearly, e.g. in the caption of fig. 1.

Referee #1 Comments:

I have read this paper from the perspective of a reader who is not familiar with input-output analysis and the concept of footprinting/embodied impacts. For me there are two pressing issues with the current paper. Firstly, the results section appears unclear to me and I cannot follow what is actually presented, and secondly, I believe that the authors should provide more context for the results. Currently, there is a focus on the actual numerical results, but contextual information about how the numerical results can be interpreted (through examples or specific insights). More explanation will help the reader to understand the meaning of the results much better. Further, I think the paper is narrow in its exploitation of the topic and the results. Please see me detailed comments below.

Response: We thank the referee for the constructive comments. We have revised the manuscript extensively to set our study in a broader context.

Abstract. The key feature of this article is not made clear: This paper will provide unprecedented quantitative insight into the supply chain patterns that link final consumption to premature deaths. I think the fact that the authors are able to account premature deaths to consumption that is geographically removed from where the death occurs must be developed further. Further, the authors use the word *cause* to link final consumers to the premature deaths. I would recommend to use - throughout the article – different vocabulary for the actual cause of death (which is always local), and the underlying driver (for example final consumption of a good far removed from the place where the death occurs). For example, the cause of death could be air pollution, which is accounted to final consumption somewhere else. In my opinion, sentences such as the richest 10% of Chinese consumers cause 15% of all premature air pollution deaths can easily be misinterpreted.

Response: Again, we appreciate these comments and have worked to revise the text accordingly. Specifically, we emphasize more strongly the key insight pointed by the Referee (e.g., lines 68-76), and we no longer use the word “cause” to refer to deaths resulting from emissions embodied in consumed goods and services (i.e. “indirect emissions”); rather we use terms such as “related to” and “account for.”

Line 58. This is in my opinion the correct use of the word *cause*, because air pollution is identified as the actual cause of the deaths.

Response: We agree and have left the word in this instance.

Line 63, This is a key sentence in the article: direct impacts and embodied impacts are explained. I think the authors should stick to their introduced terms throughout the entire paper. I will use the words direct and embodied throughout this review. It may also be advisable to add one or two more sentences to clarify the concept of embodied air pollution for readers that are not familiar with it.

Response: We have added text to this paragraph that describes in greater detail the related literature, what we have done, defines embodied emissions, and introduces terminology that we use consistently throughout: “direct” and “indirect”:

“Yet household consumption may also indirectly impact human health via air pollution virtually embodied in goods and services consumed¹⁴⁻¹⁶, and regional clean air policies which focus on direct sources may thus encourage “leakage” of polluting activities to other regions which may have less resources to control emissions and provide health services¹⁷. Whereas previous studies have examined the embodied water use^{18,19}, energy consumption^{20,21}, and emissions²²⁻²⁴ of regions’ household consumption (including the roles of income, geography, culture, age, household size, regional policies²⁵⁻²⁷), no previous studies have quantified air pollution-related deaths embodied in household consumption. In addition to linking the locations of consumed goods to sources of air pollution, such a consumption-based accounting of air pollution deaths also requires tracking the physical transport of that pollution in the atmosphere and estimating the related deaths. Herein, we distinguish air pollution deaths related to emissions directly produced by a household from those related to emissions embodied in goods consumed by a household as “direct” and “indirect”, respectively.”

References:

- 14 Huo, H. *et al.* Examining air pollution in China using production-and consumption-based emissions accounting approaches. *Environmental science & technology* **48**, 14139-14147 (2014).
- 15 Guan, D. *et al.* The socioeconomic drivers of China’s primary PM_{2.5} emissions. *Environmental Research Letters* **9**, 024010 (2014).
- 16 Zhao, H. *et al.* Assessment of China’s virtual air pollution transport embodied in trade by using a consumption-based emission inventory. *Atmospheric Chemistry and Physics* **15**, 5443-5456 (2015).
- 17 Feng, K. *et al.* Outsourcing CO₂ within China. *Proceedings of the National Academy of Sciences* **110**, 11654-11659 (2013).
- 18 Cai, B., Liu, B. & Zhang, B. Evolution of Chinese urban household’s water footprint. *Journal of Cleaner Production* **208**, 1-10 (2019).
- 19 Guan, D. *et al.* Lifting China’s water spell. *Environmental science & technology* **48**, 11048-11056 (2014).
- 20 Golley, J. & Meng, X. Income inequality and carbon dioxide emissions: the case of Chinese urban households. *Energy Economics* **34**, 1864-1872 (2012).
- 21 Ding, Q., Cai, W., Wang, C. & Sanwal, M. The relationships between household consumption activities and energy consumption in china—an input-output analysis from the lifestyle perspective. *Applied energy* **207**, 520-532 (2017).
- 22 Liu, L.-C., Wu, G., Wang, J.-N. & Wei, Y.-M. China’s carbon emissions from urban and rural households during 1992–2007. *Journal of Cleaner Production* **19**, 1754-1762 (2011).
- 23 Zhang, J., Yu, B., Cai, J. & Wei, Y.-M. Impacts of household income change on CO₂ emissions: An empirical analysis of China. *Journal of Cleaner Production* **157**, 190-200 (2017).
- 24 Wiedenhofer, D. *et al.* Unequal household carbon footprints in China. *Nature Climate Change* **7**, 75 (2016).
- 25 Ivanova, D. *et al.* Mapping the carbon footprint of EU regions. *Environmental Research Letters* **12**, 054013 (2017).

26 Jones, C. M. & Kammen, D. M. Quantifying carbon footprint reduction opportunities for US households and communities. *Environmental science & technology* **45**, 4088-4095 (2011).
27 Shigetomi, Y., Nansai, K., Kagawa, S. & Tohno, S. Trends in Japanese households' critical-metals material footprints. *Ecological Economics* **119**, 118-126 (2015).

Line 70. Here the authors use the word related. I assume you mean embodied?

Response: We have quantified the deaths related to both direct and indirect (i.e. embodied) emissions. We have rephrased the sentence to make it clear: "...we quantify the air pollution health impacts from both direct and indirect emissions of household consumption for 12 income groups (5 for rural and 7 for urban) over 30 provinces in mainland China."

Line 75. Here the authors use the word caused for what I understand are in fact embodied pollutant emissions.

Response: As mentioned above, the impacts of both direct and indirect emissions were estimated. We have rephrased the sentence as follows: "In summary, we use a detailed inventory for anthropogenic emissions in China (MEIC: <http://www.meicmodel.org/>), a survey-based inventory of direct emissions from rural households, household consumption statistics, and a multi-regional input-output model of the Chinese economy to quantify direct and indirect pollutant emissions from household daily consumption by various income groups..."

Line 81. From here onwards, I do get confused whether direct or embodied deaths are discussed. Are emissions also part of the remaining 11% of death causes? Please give some examples what the natural causes are. It would also be interesting to see how the deaths by natural causes per 1 million inhabitants compare to other nations or for example the OECD average. I assume that from here onwards all assessment is done for the 89% of deaths that are accounted to anthropogenic emissions, and the remaining 11% are not further discussed in this paper? Please clarify.

Response: The referee is correct that the remaining 11% are not discussed further in this paper. Those deaths were related to natural sources such as mineral dust, forest fires, and biogenic emissions. In the revised manuscript, we have removed the statement regarding natural sources since this is not within the scope of the work. Figure 1A is also removed from the revised paper to avoid misleading.

Line 85. Those 42% of the 89% of all deaths that are accounted to household consumption, is that embodied deaths, or are they caused by direct exposure such as fuel burning that you discussed earlier? The pie chart Fig 1 B lists the standard IO final demand categories, plus international transport. I assume you split the household consumption into rural and urban, which would make these 42% purely embodied. My understanding is that there is a strong proportion of non trade-related deaths through direct combustion of fuels. These cannot be included in this calculation of embodied deaths, correct? Please clarify, what you are discussing here.

Response: Those 44% deaths include both direct emissions (20%) and embodied/indirect emissions (24%). In original Fig. 1B, premature deaths related to household consumption include both direct impact and embodied deaths. In the revised manuscript, we have combined Fig. 1B and 1C to one figure and rephrased the related statement to improve the clarity.

Line 87. Here you explain that a proportion of those deaths are due to direct impacts. I assume this means that the intermediate transaction part of the satellite block of your MRIO model receives a certain percentage of those 89% of anthropogenically-driven deaths, and the remainder are assigned to final household consumption, correct? Then Fig 1 B must be a mix of direct and embodied deaths? Please clarify.

Response: Yes, as mentioned above, original Fig. 1B include premature deaths related to both direct and embodied/indirect deaths due to household consumption. For now, we used a single pie chart to represent Fig. 1B and 1C to improve the clarity.

Fig 2. From a first view, it looks like the majority of deaths occur in very rich families. I do not understand the bars for each income class. Firstly, I would recommend putting all the rural statistics to the left of the chart, and urban statistics to the right and then sort both the urban and the rural bars according to income class. This might reveal trends that are currently not easy to spot. Secondly, each bar shows a *direct consumption and indirect consumption*, but the y-axis shows the deaths per consumer (I assume this is direct and embodied combined?). Does each bar show the directly suffered deaths, and the embodied deaths in each class that are suffered, or that are caused? The black dotted line shows the per capita income? This is not clear from the legend.

Response: As suggested by the referee, we now separate rural statistics from urban statistics and ordered them according to their income level respectively. The y-axis represents the per capita total air pollution-related deaths from both direct and indirect emissions that are related to each groups' consumption, which should be 'caused' instead of 'suffered'. The dotted line show the per capita income of each group. We now made these clear in the figure caption.

Line 101. The sentence *For example, deaths due to direct emissions of extremely poor rural consumers were 22% greater those caused by direct emissions of high-income rural consumers* should be linked to the statement you made before. At the moment, most of the statements in the Results section stand by themselves, including this one.

Response: We have revised our manuscript to stick to our point. "For direct emissions, poor rural residents are related to more deaths than richer rural residents as poorer households tend to consume more solid fuel than richer households. Deaths due to direct emissions of extremely poor rural residents are 20% greater than those due to direct emissions of high-income rural residents (Fig. 2), and the poorest 10% of rural residents are related to 21% of air pollution-related deaths from direct emissions (see Fig. 3C)."

Line 137. Here you are switching your focus towards the Lorenz Curve. How does that fit into the storyline of this section?

Response: The Lorenz Curve is a supplement of Figure 2 that helps the readers to further understand the inequality of air pollution-related deaths due to consumption and income earned by Chinese households in 2012. We now make it as the new Figure 3. From the new Figure 3, we could learn that the richest 10% of residents earn 29% of total household income and link to 13% of air pollution-related deaths. In comparison, the poorest 10% of residents earn only 1% of total household income yet link to 11% of air pollution-related deaths. While the distribution of indirect impacts is consistent with the income (Fig. 3C), the impacts from direct emissions are inversely distributed to income level (Fig. 3D), meaning that air pollution-related deaths are more evenly distributed than income earned nationwide: whereas the Gini coefficient in 2012 was 0.418 (Fig. 3B), the analogous inequality coefficient of air pollution-related deaths in the same year was only 0.014 (Fig. 3A).

Fig 3. This entire figure needs to be revised. First of all, I would recommend that you motivate that you are taking a different angle in this part of the results section. You are now presenting spatial results. What is the y-dimension of this axis? Part A of the figure is clear (apart from the y axis). In part B it is unclear whether direct or embodied deaths are reported, and whether the deaths are driven or suffered. Part C's title Deaths occurred in other regions does not fit with the rest of the study. Are these embodied deaths in the final consumption of each respective region? The same points hold for Part D. If the y axis are in fact the region listed on the right hand of the graph, then it does not make sense to connect the dots with a line in parts A, C, and D.

Response: We have revised the original Figure 3 and made it as the new Figure 4. And as suggested by the referee, this figure aims at showing the spatial differences in air pollution-related deaths related to household consumption in seven different regions of China. The y-axis represents the seven regions, which is now clearly marked. We also separate the direct and indirect deaths, which are all deaths “driven” by each region instead of “suffered”. Figure 4B now compares the fraction of deaths occurred in other places due to each region’s direct and indirect consumption. Figure 4C shows the transfers of deaths between regions from both direct and indirect perspective. We also revised the related paragraph (e.g. line 130-152).

Fig 4. What is the take away message from Figure 4? I can see that low income households suffer higher direct health impacts, but you already showed that before. I suggest you discuss how Part A, C, and D fit together, especially since Part D might appear surprising, because poorer households suffer less than richer ones in this curve. All in all I do question if Fig 4 offers any more insights.

Response: As mentioned above, Figure 4 (i.e. the new Figure 3) is a supplement of Figure 2 that further illustrates the inequality of air pollution-related deaths due to consumption and income earned by Chinese households in 2012. We find that while the distribution of indirect impacts is consistent with the income, the impacts from direct emissions are inversely

distributed to income level. As a result, the total air pollution-related deaths are more evenly distributed than income earned nationwide: whereas the Gini coefficient in 2012 was 0.418, the analogous inequality coefficient of air pollution-related deaths in the same year was only 0.014.

Remainder of the Results section. The remainder of this section appears also as a compilation of data observations, but they lack a consistent storyline that connect the presented results. Also, as mentioned at the beginning of this review, the different statements should receive some contextual framing to highlight how the findings translate into the real world.

Response: We have revised our manuscript as suggested by the referee.

Discussions and Conclusions. In this section I would expect the key take-away message from this paper. The fact that income is related to impact has been shown countless times before, and does not qualify as a new insight, despite the topical nature of this study. Additionally, too many of the statements made in this section are speculative or lack evidence.

Response: We have rewritten the *Discussion and Conclusions* section to emphasize our new findings and avoid speculative statements. In this work, we developed the quantitative relationship between household consumption and air pollution-related premature deaths for the first time, including both direct impact from solid fuel use and indirect impact embodied in goods and services consumed. Although income is generally related to impact, we find unexpected higher contribution of rural household consumption to air pollution-related deaths in China because impacts from direct emissions in rural area are not positively related to income level. Our work also provides unprecedented quantitative insight into the supply chain patterns that link final consumption to air-pollution related premature deaths, which offers a basis for clean air policies that avoid and redress socio-economic and regional inequities.

Sentence starting in line 158. Please rephrase and explain what you mean by complicated.

Response: Here we mean we find unexpected higher contribution of rural household consumption to air pollution-related deaths in China, largely due to direct emissions from solid fuel (including coal and biomass) combustion in rural China. We have revised our sentence.

Sentence starting in line 168. I think this statement can also be derived without your study. You mentioned that a large number of deaths are caused by exposure to local fuel combustion. Reducing combustion of these fuels will therefore inevitably lead to reduction in deaths.

Response: We thank the referee for the constructive comment. In response to this comment, the section of *Discussion and Conclusions* has been rewritten in the revised version. The related discussion on direct fuel consumption has been revised as follows:

“Although substantial contribution of solid fuel use on air pollution in China has been investigated³⁵⁻³⁷, we find unexpected higher contribution of rural household consumption to air pollution-related deaths in China. These findings further emphasize the great importance of mitigating emissions from direct emissions of rural households, given that current policies

focus more on urban pollution. Indeed, our results likely underestimate air pollution-related deaths by rural households because neglecting the impact of indoor air pollution from solid fuel use. Policies that promote clean energy (e.g., natural gas and electricity use) in rural households could provide a perfect solution, however, the high prices, lack of accessibilities to natural gas, and traditional consumption behavior might hinder the promotion of such policy³⁸. Because urban residents also suffer the pollution from rural emissions and have higher willingness to pay for alleviating pollution, providing price subsidy within a certain time period might be a possible solution given that urban residents pay more taxes than rural residents. The price of clean energy will be eventually accepted by rural residents with economy developed and income increased.”

Line 177. I believe that you cannot discuss urbanisation in relation to the discussed health impacts without considering other factors such as transitioning to other fuels etc. This analysis as it stands is incomplete.

Response: We have removed this statement in the revised manuscript.

References. A lot of work has been done on sub-national MRIO and also pollution. However, the authors do not discuss their results in the light of previous works in great detail. I recommend that the authors undertake a more thorough literature review on the topic.

Response: We have added the following discussion in the revised manuscript.

“Yet household consumption may also indirectly impact human health via air pollution virtually embodied in goods and services consumed¹⁴⁻¹⁶, and regional clean air policies which focus on direct sources may thus encourage “leakage” of polluting activities to other regions which may have less resources to control emissions and provide health services¹⁷. Whereas previous studies have examined the embodied water use^{18,19}, energy consumption^{20,21}, and emissions²²⁻²⁴ of regions’ household consumption (including the roles of income, geography, culture, age, household size, regional policies²⁵⁻²⁷), no previous studies have quantified air pollution-related deaths embodied in household consumption.”

“This is within the line of other studies related to income and environmental impact; impact of rich urban consumers are higher than poor consumers by a factor of 3.8–9.5 for different environmental indicators (i.e., CO₂ emissions, air and water pollutant emissions, and water use)^{18,23,24,39}. Our work provides additional insight to this discussion by adding air pollution-related premature deaths as a new indicator.”

References:

14 Huo, H. *et al.* Examining air pollution in China using production-and consumption-based emissions accounting approaches. *Environmental science & technology* **48**, 14139-14147 (2014).

15 Guan, D. *et al.* The socioeconomic drivers of China’s primary PM_{2.5} emissions. *Environmental Research Letters* **9**, 024010 (2014).

- 16 Zhao, H. *et al.* Assessment of China's virtual air pollution transport embodied in trade by using a consumption-based emission inventory. *Atmospheric Chemistry and Physics* **15**, 5443-5456 (2015).
- 17 Feng, K. *et al.* Outsourcing CO₂ within China. *Proceedings of the National Academy of Sciences* **110**, 11654-11659 (2013).
- 18 Cai, B., Liu, B. & Zhang, B. Evolution of Chinese urban household's water footprint. *Journal of Cleaner Production* **208**, 1-10 (2019).
- 19 Guan, D. *et al.* Lifting China's water spell. *Environmental science & technology* **48**, 11048-11056 (2014).
- 20 Golley, J. & Meng, X. Income inequality and carbon dioxide emissions: the case of Chinese urban households. *Energy Economics* **34**, 1864-1872 (2012).
- 21 Ding, Q., Cai, W., Wang, C. & Sanwal, M. The relationships between household consumption activities and energy consumption in china—an input-output analysis from the lifestyle perspective. *Applied energy* **207**, 520-532 (2017).
- 22 Liu, L.-C., Wu, G., Wang, J.-N. & Wei, Y.-M. China's carbon emissions from urban and rural households during 1992–2007. *Journal of Cleaner Production* **19**, 1754-1762 (2011).
- 23 Zhang, J., Yu, B., Cai, J. & Wei, Y.-M. Impacts of household income change on CO₂ emissions: An empirical analysis of China. *Journal of Cleaner Production* **157**, 190-200 (2017).
- 24 Wiedenhofer, D. *et al.* Unequal household carbon footprints in China. *Nature Climate Change* **7**, 75 (2016).
- 25 Ivanova, D. *et al.* Mapping the carbon footprint of EU regions. *Environmental Research Letters* **12**, 054013 (2017).
- 26 Jones, C. M. & Kammen, D. M. Quantifying carbon footprint reduction opportunities for US households and communities. *Environmental science & technology* **45**, 4088-4095 (2011).
- 27 Shigetomi, Y., Nansai, K., Kagawa, S. & Tohno, S. Trends in Japanese households' critical-metals material footprints. *Ecological Economics* **119**, 118-126 (2015).
- 39 Liu, L.-C. & Wu, G. Relating five bounded environmental problems to China's household consumption in 2011–2015. *Energy* **57**, 427-433 (2013).

Referee #2 Comments:

General comment on Main text:

The main text is well-written and potentially interesting, however I found a number of unclear points for the analysis used in this study. I like the story line of the paper on how the pollution inequality has been occurred in China? On the other hand, the analysis was based on some important assumptions on the model analysis and data. I am wondering whether or not the discussion of the paper is well supported by the reliable quantitative analysis. For example, the proper sector correspondence between household expenditure survey data and MRIO data is crucial in estimating the sectoral and provincial PM_{2.5} emissions and health impacts. In addition, the household expenditure survey and income data should be well constructed in the Chinese provinces. Yes, this is the first attempt to do it and assumptions are necessary in doing it. The step-by-step analysis based on the reliable consumption statistics would be important in the valuable quantitative analysis. I would like to admire the authors in this ambitious study, however I am negative in publishing this paper in the Nature Communications. A more detailed household consumption and income survey data consistent with the MRIO data is crucial in analyzing the income and pollution inequalities.

Response: We thank the referee for the valuable comments. We agree that the well-established household income and expenditure data and the proper sector mapping between household expenditure data and MRIO data is crucial in estimating the sectoral and provincial emissions and health impacts. We have tried our best to minimize the uncertainties induced by the absence of complete expenditure data at provincial level and the mapping process between expenditure and MRIO data. First, we have obtained the best available household expenditure survey and income data in China, although the data are still incomplete for some provinces and we have to make certain assumptions. These assumptions have also been used in previous studies (e.g. Wiedenhofer et al., 2017). Second, we followed the standard mapping process that established and widely used in previous studies (e.g. Liu et al., 2011; Wiedenhofer et al., 2017; Zhang et al., 2017). Third, we improved our mapping method in the revised version, to reduce the uncertainties brought by sector mapping (Please see the detailed response below). Last but not least, we quantify and discuss the uncertainties for each of the steps, provide the overall ranges of uncertainties (95%CI) of the main results, and found that our results are quite robust to different assumptions (Please see the detailed response below).

Therefore, we believe that the model and data used in this study are reliable and could well support our conclusions. We also revised our *Supplementary Information* extensively to more clearly describe all the models, methods and uncertainty analyses in our study.

References:

- Liu, L.-C., Wu, G., Wang, J.-N. & Wei, Y.-M. China's carbon emissions from urban and rural households during 1992–2007. *Journal of Cleaner Production* **19**, 1754-1762 (2011).
- Wiedenhofer, D. *et al.* Unequal household carbon footprints in China. *Nature Climate Change* **7**, 75 (2016).
- Zhang, J., Yu, B., Cai, J. & Wei, Y.-M. Impacts of household income change on CO₂ emissions: An empirical analysis of China. *Journal of Cleaner Production* **157**, 190-200 (2017).

Specific comment on Supplementary Information:

(1) Page 3, Line 69: Geng et al. (2015) provided the result of spatial distribution of the satellite-derived PM_{2.5} concentrations in China for 2006–2012. What result did you use in this study? What year? The inventory database is available to the public?

Response: In this study, we used the annual mean satellite-based PM_{2.5} concentrations in 2012 for the estimates of PM_{2.5}-related premature deaths in 2012. The datasets can be obtained upon request to the corresponding author of Geng et al. (2015).

(2) Page 3, Line 73: The IER model was used to convert the PM_{2.5} concentrations into PM_{2.5}-related premature deaths. What are the values of the parameters, alpha, gamma, and delta? Please clarify them.

Response: In the IER function, α , γ , and δ are parameters used to describe the shape of the concentration-response curve, and C_0 is the counterfactual concentration. Burnett et al. (2014) provided 1,000 sets of these parameters for the estimates of the median value and the uncertainty range. Since the GEOS-Chem adjoint model requires a differentiable equation as its cost function, Lee et al. (2015) fitted IER curves to the median of the suite of IER functions used in the GBD project. Specific parameters and graphs of the curves can be found in SI section S-6 of Lee et al. (2015).

In this study, we adopted the fitted parameters from Lee et al. (2015) for the estimation of PM_{2.5}-related premature deaths and as cost function in the GEOS-Chem adjoint model. The value of α , γ , δ and C_0 are now listed in Supplementary Table 2, as shown below.

Supplementary Table 2. Parameters and death incidences used in IER model.

	IHD	Stroke	COPD	LC	Source
α	0.843	1.01	18.3	159	Lee et.al., 2015
γ	0.0724	0.0164	0.000932	0.000119	Lee et.al., 2015
δ	0.544	1.14	0.682	0.735	Lee et.al., 2015
C_0	6.96	8.38	7.17	7.24	Lee et.al., 2015
B	0.000707	0.00129	0.000696	0.000383	Forouzanfar et al., 2015

References:

Burnett, R. T. *et al.* An integrated risk function for estimating the global burden of disease attributable to ambient fine particulate matter exposure. *Environ. Health Perspect.* **122**, 397-403 (2014).

Lee, C. J. *et al.* Response of global particulate-matter-related mortality to changes in local precursor emissions. *Environ. Sci. Technol.* **49**, 4335-4344 (2015).

(3) Page 3, Line 89: Why you did not use the population database for 2012 that is consistent with the data of the annual mean PM_{2.5} concentrations in 2012?

Response: It was a typo here. We used the population database for 2012 in our study.

(4) Page 3, Line 93: How did you set the value of non-anthropogenic emissions, C_{no_anth}? The anthropogenic emissions can be calculated by the equation, C-C_{no_anth}. Then inserting the value of C-C_{no_anth} into the right-hand side of equation (1), C provides the relative risk based on the anthropogenic emissions, RR_{anth}. Eq. (1) is a non-linear equation, whereas Eq. (3) is a linear equation. It seems to me that Eq. (3) is something wrong, and you can straightforwardly use the non-linear equation of (1). Please clarify it.

Response: In this study, we used the direct proportion approach, which assumed a linear relationship between the proportion of total PM_{2.5} concentration to the proportion of total mortality, to estimate the source-specific premature deaths. The scientific basis of this assumption has been validated by a GBD research, GBD MAPS (GBD MAPS Working Group, 2016). In general, due to the nonlinearity of the IER functions, the source-specific contribution to PM_{2.5} mortality is non-uniform along the curve. Each source contributes a certain proportion to total PM_{2.5} concentration, but we could not assume the contribution is located in distinctive intervals of concentration. The direct proportion approach is insensitive to the order in which each source is removed from the total concentration. It also has the advantage that the sum of mortality estimates from all sources equals the mortalities from ambient PM_{2.5} exposure. Previous studies have used the direct proportion approach to solve the non-linear problem (Chafe et al., 2014; Zhang et al., 2017).

In Eq. 3 (now Eq. 18), we used the proportion of PM_{2.5} concentrations from anthropogenic sources to the total PM_{2.5} concentrations ($(C_{all} - C_{(no_anth)})/C_{all}$) to estimate the deaths related to anthropogenic PM_{2.5}. We added the following sentence in the revised *Supplementary Information* to clarify the methods we used:

“Here, we used the direct proportion approach, which assumed a linear relationship between the proportion of total PM_{2.5} concentration to the proportion of total mortality, to estimate the source-specific premature deaths. The scientific basis of this assumption has been validated by a GBD research, GBD MAPS (GBD MAPS Working Group, 2016).”

References:

Chafe, Z. A. *et al.* Household cooking with solid fuels contributes to ambient PM_{2.5} air pollution and the burden of disease. *Environ. Health Perspect.* **122**, 1314-1320 (2014).

GBD MAPS Working Group. Burden of Disease Attributable to Coal-Burning and Other Major Sources of Air Pollution in China. Special Report 20. Boston, MA:Health Effects Institute (2016).

Zhang, Q. *et al.* Transboundary health impacts of transported global air pollution and international trade. *Nature* **543**, 705-709 (2017).

(5) Page 4, Line 108: What is $E_{i,j,k}$? M is the function of the E ? I see that a combined equation from three equations, (1), (2), and (3) includes some inconsistencies in the mathematical model. A critical assumption may be included there. I don't know about it.

Response: In our study, we use the GEOS-Chem adjoint model to link emissions to premature deaths. The adjoint model is able to determine the response of $PM_{2.5}$ -related mortality to changes in input emissions. Following Lee et al. (2015), we define the cost function in the adjoint model as the anthropogenic $PM_{2.5}$ related premature deaths resulting from long-term exposure to $PM_{2.5}$ (M). The outputs provided by the adjoint model ($\frac{\partial M}{\partial E_{i,j,k}}$) are the partial derivatives of this cost function (M) with respect to anthropogenic emissions ($E_{i,j,k}$) in the simulation domain, which we refer to as the sensitivities of receptor region's $PM_{2.5}$ related premature deaths to emissions at all species, locations and times.

(6) Page 4, Line 134-Page 5, Line 135: The data on household direct emissions is available to the public? What is the MEIC model? The authors say that 'these emissions attend to occurred locally.' What is the meaning of this sentence? Why locally? The emissions can be widely spread by the wind, therefore the atmospheric transport model can work.

Response: The household emission inventory is developed by our group (Peng et al., 2019). Emissions from the rural residential sector are estimated using survey-based solid fuel use data and improved emission factors (Peng et al., 2019). In summary, a generalized additive model (GAM) is developed to estimate rural residential solid fuel consumption at county-level using first-hand data collected by a questionnaire survey. Then atmospheric pollutant emission inventory was calculated using the estimated solid fuel use data and an improved emission factor database.

The MEIC model (Multi-resolution Emission Inventory for China; <http://www.meicmodel.org>), which is developed and maintained by Tsinghua University, provides the anthropogenic emissions of sulfur dioxide (SO_2), nitrogen oxides (NO_x), ammonia (NH_3), black carbon (BC), organic carbon (OC) and anthropogenic $PM_{2.5}$ dust used in this study. The MEIC is a bottom-up emission inventory framework which covers 31 provinces in mainland China and includes more than 700 anthropogenic emitting sources. It is improved based on the bottom-up emission inventories developed by the same group (Zhang et al., 2009), which use technology and process based methods to resolve the quantitative relationship between emissions and technology turnover. Detailed description of the technology-based methodology and the source classifications can be found elsewhere (Zhang et al., 2007, 2009; Lei et al., 2011; Li et al., 2017). The sources of the underlying data used in the MEIC model, including the activity rates, technology penetration data and emission factors, are summarized in Zheng et al. (2018). We have added the description of the MEIC model in the revised *Supplementary Information (S1)*. The sentence “*these emissions attend to occurred locally*” is relative to those of the indirect emissions, which are produced and emitted throughout the supply chains among sectors and regions who take part in the production process of household consumed goods or services. To be more precisely, we have rephrased this sentence to “All direct emissions are emitted locally”.

References:

- Lei, Y., Zhang, Q., He, K. B. & Streets, D. G. Primary anthropogenic aerosol emission trends for China, 1990–2005. *Atmos. Chem. Phys.* **11**, 931-954 (2011).
- Li, M. *et al.* MIX: a mosaic Asian anthropogenic emission inventory under the international collaboration framework of the MICS-Asia and HTAP. *Atmos. Chem. Phys.* **17**, 935-963 (2017).
- Peng, L. *et al.* Underreported coal in statistics: A survey-based solid fuel consumption and emission inventory for the rural residential sector in China. *Applied Energy* **235**, 1169-1182 (2019).
- Zheng, B. *et al.* Trends in China's anthropogenic emissions since 2010 as the consequence of clean air actions. *Atmos. Chem. Phys. Discuss.* **2018**, 1-27 (2018).
- Zhang, Q. *et al.* NO_x emission trends for China, 1995–2004: The view from the ground and the view from space. *J. Geophys. Res-Atmos.* **112**, doi:doi:10.1029/2007JD008684 (2007).
- Zhang, Q. *et al.* Asian emissions in 2006 for the NASA INTEX-B mission. *Atmos. Chem. Phys.* **9**, 5131-5153 (2009).

(7) Page 5, Eq. (6): t can be subscript.

Response: Revised as suggested.

(8) Page 5, Lines 155-158: e is important. What is the database of e? Is it available to the public? What is the mapping process? Equation (9) provides the region-specific pollutant emissions embedded in final demand of China and exports.

Response: Yes, **e** is important in our study since it is the region-specific pollutant emission used to produce \hat{f} . The database of **e** are from the MEIC model, a published and publicly available model, which covers 31 provinces in mainland China and includes more than 700 anthropogenic emitting sources, as mentioned above and in section S1 in the revised *Supplementary Information*.

The detailed mapping process between the MEIC model and the MRIO model can be found in our previous studies (Huo et al., 2014; Zhao et al., 2015). We aggregated the 700+ emission sources and production categories into 46 production sectors based on the classification method of China's official energy statistical yearbook. Detailed mapping tables are shown below:

ID	Sector in MRIO	ID	Sector in Emission Inventory
1	Agriculture	1	Farming, Forestry, Animal Husbandry, Fishery & Water Conservancy
2	Coal mining and processing	2	Coal Mining and Dressing
3	Crude petroleum and natural gas products	3	Petroleum and Natural Gas Extraction
4	Metal mining	4	Ferrous Metals Mining and Dressing
		5	Nonferrous Metals Mining and Dressing
5	Nonmetal mining	6	Nonmetal Minerals Mining and Dressing
		7	Other Minerals Mining and Dressing
6		8	Food Processing
		9	Food Production

	Manufacture of food products and tobacco processing	10	Beverage Production
		11	Tobacco Processing
7	Textile	12	Textile Industry
8	Clothing, leather, fur, etc.	13	Garments and Other Fiber Products
		14	Leather, Furs, Down and Related Products
9	Wood processing and furnishing	15	Timber Processing, Bamboo, Cane, Palm & Straw Products
		16	Furniture Manufacturing
10	Paper making, printing, stationery, etc.	17	Papermaking and Paper Products
		18	Printing and Record Medium Reproduction
		19	Cultural, Educational and Sports Articles
11	Petroleum refining, coking, etc.	20	Petroleum Processing and Coking
12	Chemical industry	21	Raw Chemical Materials and Chemical Products
		22	Medical and Pharmaceutical Products
		23	Chemical Fiber
		24	Rubber Products
		25	Plastic Products
13	Nonmetal products	26	Nonmetal Mineral Products
14	Metallurgy	27	Smelting and Pressing of Ferrous Metals
		28	Smelting and Pressing of Nonferrous Metals
15	Metal products	29	Metal Products
16	General and specialist machinery	30	Ordinary Machinery
		31	Equipment for Special Purpose
17	Transport equipment	32	Transportation Equipment
18	Electrical equipment	33	Electric Equipment and Machinery
19	Electronic equipment	34	Electronic and Telecommunications Equipment
20	Instrument and meter	35	Instruments, Meters Cultural and Office Machinery
21	Other manufacturing	36	Other Manufacturing Industry
22	Electricity and hot water production and supply	37	Electric Power, Steam and Hot Water Production and Supply
23	Gas and water production and supply	38	Gas Production and Supply
		39	Tap Water Production and Supply
24	Construction	40	Construction
25	Transport and storage	41	Transport, Storage, Postal & Telecommunications Services
26	Wholesale and retailing	42	Wholesale, Retail Trade and Catering Service
27	Hotel and restaurant		
28	Leasing and commercial services	43	Recycling and Disposal of Waste Other
29	Scientific research		
30	Other services		
		45	Urban residential consumption
		46	Rural residential consumption

References:

- Huo, H. *et al.* Examining Air Pollution in China Using Production- And Consumption-Based Emissions Accounting Approaches. *Environ. Sci. Technol.* **48**, 14139-14147 (2014).
- Zhao, H. Y. *et al.* Assessment of China's virtual air pollution transport embodied in trade by using a consumption-based emission inventory. *Atmos. Chem. Phys.* **15**, 5443-5456 (2015).

(9) Page 5, Line160: Equation (10) provides the region-specific pollutant emissions embedded in final demand category t of region s . Specifically, Eq. (10) captures that the household consumption of Beijing can induce pollutant emissions in Jilin. This emission modeling technique is NOT new and many studies used this footprint analysis. The pollutant emissions from Jilin embedded in Beijing can be used as the database of estimating PM_{2.5} related premature death by inserting the embedded PM_{2.5} emissions into the right-hand side of Eq. (1)? I am wondering how the footprint analysis was combined with the risk impact analysis. There is no explanation about the mathematical connection. The novelty is unclear!

Response: The referee is right that previous studies have used similar approach to estimate the CO₂ (Diana *et al.*, 2017; Wiedenhofer *et al.*, 2017) and air pollutant (Zhao *et al.*, 2015; Zhao *et al.*, 2017) emissions embodied in trade flow. However, in our study, we went a few steps further, linking the emissions to health risks and tracing the health risks to different income groups by involving the GEOS-Chem adjoint model, the IER function and the income data. Although these models are not new, our work is the first attempt to combine them all together and estimate the inequality of household consumption and air pollution deaths in China.

The conversions from air pollutant emissions to PM_{2.5} related premature deaths are achieved by the GEOS-Chem adjoint model. The adjoint of GEOS-Chem is able to determine the response of PM_{2.5}-related mortality to changes in emissions of inorganic precursor gases (i.e. SO₂, NO_x and NH₃), carbonaceous particles (i.e. BC and OC) and primary anthropogenic PM_{2.5} dust. It allows for efficient computation of the partial derivatives of a scalar model response with respect to input conditions (e.g. emission rates). Previous studies have used the adjoint of GEOS-Chem to quantify the response of PM_{2.5} concentrations and air pollution mortality to emissions sources (Pappin *et al.*, 2013; Lee *et al.*, 2015; Zhang *et al.*, 2015). In our study, we define the cost function in the adjoint model as the anthropogenic PM_{2.5} related premature deaths resulting from long-term exposure to PM_{2.5}. Then the model outputs the partial derivatives of this cost function with respect to anthropogenic emissions in the simulation domain, which we refer to as the sensitivities of receptor region's PM_{2.5} related premature deaths to emissions at all species, locations and times. Combining the sensitivity simulated by the GEOS-Chem adjoint model and the urban or rural household emissions by each income group, we could attribute the PM_{2.5} related deaths to each income group.

References:

- Diana, I. *et al.* Mapping the carbon footprint of EU regions. *Environ.Res. Lett.* **12**, 054013 (2017).
- Lee, C. J. *et al.* Response of global particulate-matter-related mortality to changes in local precursor emissions. *Environ. Sci. Technol.* **49**, 4335-4344 (2015).

Pappin, A. J. & Hakami, A. Source attribution of health benefits from air pollution abatement in Canada and the United States: an adjoint sensitivity analysis. *Environ. Health Perspect.* **121**, 572 (2013).

Wiedenhofer, D. *et al.* Unequal household carbon footprints in China. *Nature Clim. Change* **7**, 75-80 (2017).

Zhao, H. Y. *et al.* Assessment of China's virtual air pollution transport embodied in trade by using a consumption-based emission inventory. *Atmos. Chem. Phys.* **15**, 5443-5456 (2015).

Zhao, H. *et al.* Effects of atmospheric transport and trade on air pollution mortality in China. *Atmos. Chem. Phys.* **17**, 10367-10381 (2017).

Zhang, L. *et al.* Source attribution of particulate matter pollution over North China with the adjoint method. *Environ. Res. Lett.* **10**, 084011 (2015).

(10) Page 6, Line 166: What is 'sum'? Sigma_s would be correct.

Response: Revised as $ce_t^s = \sum_r e_t^s + de_t^s$.

(11) Page 6, Line 180: What is the sampling survey? Did you do it in this study? Please provide the detailed information on it.

Response: The income and expenditure data used in this study are obtained from national and provincial statistic yearbooks (China Urban Life and Price Yearbook, 2012; China Statistical Yearbook 2013; China provincial Statistical Yearbook, 2013). The statistical yearbooks report average incomes and consumption expenditure patterns for different income groups in a province based on the sampling survey. The sampling survey were conducted by the National Bureau of Statistics of China, not in our study. We have added this information in the revised *Supplementary Information*.

References:

China Urban Life and Price Yearbook 2011 (National Bureau of Statistics, China Statistical Press, 2012)

China Statistical Yearbook 2013 (National Bureau of Statistics, China Statistical Press, 2013)

China provincial (Beijing, Tianjin, Hebei, Shanxi, Inner Mongolia, Liaoning, Jilin, Heilongjiang, Shanghai, Jiangsu, Zhejiang, Anhui, Fujian, Jiangxi, Shandong, Henan, Hubei, Hunan, Guangdong, Guangxi, Hainan, Chongqing, Sichuan, Guizhou, Yunnan, Shaanxi, Gansu, Qinghai, Ningxia, Xinjiang) Statistical Yearbook (National Bureau of Statistics, China Statistical Press, 2013)

(12) Page 6, Lines 183-190: I cannot understand these sentences. What are the ungrouped provinces? Why did you use the their average income and group data from their neighboring provinces? Is this assumption plausible? The detailed explanation would be important to readers. How the 'average' resident number of each income group was estimated? A simple calculus about it would be useful for readers. This is also unclear.

Response: The “ungrouped provinces” means provinces without available income group data. We have changed these words in the revised *Supplementary Information*.

The income and expenditure data used in this study are obtained from national and provincial statistic yearbooks (China Urban Life and Price Yearbook, 2012; China Statistical Yearbook 2013; China provincial Statistical Yearbook, 2013). However, these data are not available for all the 30 provinces considered in our study. In 2012, 90% and 60% provinces report the average incomes by groups for urban and rural households, respectively; 60% and 40% provinces report the consumption expenditure patterns by groups for urban and rural households, respectively. Therefore, we make an assumption that those provinces with no grouped income or expenditure data have similar income or expenditure **patterns** with the national average or their neighbouring provinces. For a specific province, we used the province-averaged income data adopted from the MRIO table as scale factor, and the by-group income data from the national average or the neighbouring provinces as proxies, to split the averaged income data into 12 groups.

To determine the uncertainties brought by such assumption, we conduct several sensitivity scenarios. In each scenario, we use data from the national average or one of the provinces that have available statistical data to estimate the by-group data for all the provinces with missing data. Urban and rural cases are treated separately. In total, we have 26 scenarios for urban household and 13 scenarios for rural household, as 25 and 12 provinces report income data by group for urban and rural households, respectively. We use the estimated income group data to calculate direct and indirect emissions related to each income group. The coefficient of variation (CV) of the estimated direct and indirect emissions in each income group are shown below. In general, the CV values are within $\pm 10\%$, which means the uncertainties introduced by the assumptions in our study are limited and our method is robust.

	Species	R1	R2	R3	R4	R5	U1	U2	U3	U4	U5	U6	U7
Indirect	NH ₃	0.07	0.03	0.01	0.02	0.05	0.03	0.02	0.01	0.00	0.01	0.01	0.02
	SO ₂	0.08	0.03	0.02	0.02	0.04	0.03	0.02	0.01	0.01	0.01	0.01	0.02
	NO _x	0.08	0.03	0.02	0.02	0.04	0.03	0.02	0.01	0.01	0.01	0.01	0.02
	BC	0.08	0.03	0.02	0.02	0.04	0.04	0.02	0.02	0.01	0.02	0.01	0.03
	OC	0.08	0.03	0.02	0.02	0.04	0.04	0.02	0.02	0.01	0.02	0.01	0.03
	PM _{2.5}	0.09	0.03	0.02	0.02	0.04	0.03	0.02	0.02	0.01	0.01	0.01	0.02
Direct	NH ₃	0.00	0.00	0.00	0.00	0.00	0.22	0.15	0.10	0.05	0.05	0.05	0.06
	SO ₂	0.13	0.10	0.04	0.07	0.10	0.04	0.03	0.01	0.01	0.02	0.02	0.02
	NO _x	0.02	0.01	0.01	0.01	0.02	0.05	0.05	0.03	0.02	0.02	0.03	0.04
	BC	0.02	0.02	0.01	0.02	0.03	0.04	0.03	0.02	0.01	0.02	0.02	0.03
	OC	0.01	0.01	0.00	0.01	0.01	0.04	0.03	0.02	0.01	0.02	0.02	0.03
	PM _{2.5}	0.01	0.01	0.01	0.01	0.02	0.04	0.03	0.02	0.01	0.02	0.02	0.03

The resident numbers of each groups are calculated using the following equation:

$$rn_s^j = \frac{P_s}{pph_s} \times frac_s^j \times pph_s^j$$

where rn_s^j is the resident number of income group j in province s ; P_s is the total urban or rural population in province s ; pph_s is the province-average persons per household for

province s ; pph_s^j is the average persons per household of income group j in province s ; $frac_s^j$ is the household fraction (10% or 20%) of income group j in province s . This equation has been added to the revised *Supplementary Information*.

References:

China Urban Life and Price Yearbook 2011 (National Bureau of Statistics, China Statistical Press, 2012)

China Statistical Yearbook 2013 (National Bureau of Statistics, China Statistical Press, 2013)

China provincial (Beijing, Tianjin, Hebei, Shanxi, Inner Mongolia, Liaoning, Jilin, Heilongjiang, Shanghai, Jiangsu, Zhejiang, Anhui, Fujian, Jiangxi, Shandong, Henan, Hubei, Hunan, Guangdong, Guangxi, Hainan, Chongqing, Sichuan, Guizhou, Yunnan, Shaanxi, Gansu, Qinghai, Ningxia, Xinjiang) Statistical Yearbook (National Bureau of Statistics, China Statistical Press, 2013)

(13) Page 6, Lines 195-197: What is the difference between origin group j and group i based on national grouping. Why this distinction is important in this study?

Response: There are inconsistency in the average income between the original income groups from different provinces, because regions or provinces in China experience different development stages. For example, the “poor” group of rural household in Beijing has similar average income value with the “middle” group of rural household in Heilongjiang. This might introduce biases when conducting a national analysis. Therefore, we range all income groups from 30 provinces according to their average incomes for urban and rural separately. Following the protocol by the statistic yearbooks, we allocate them into seven urban and five rural groups according to the resident numbers, which are the national grouping.

(14) Page 6, Lines 198-202: The provincial per capita household expenditure on various goods or services was used in this study. Some provinces has no statistics about it. Therefore the data of their neighboring provinces or national average was used as a proxy data. There is not the detailed explanation. This assumption is strong. Table S3 shows the sector mapping between MRIO and household consumption statistics. How did you justify the corresponding table? To me, the corresponding table of Table S3 is very rough without any justifications and sector definitions. For example, consumers hardly buy metals smelting and pressing as durable goods directly. Why residence sector is corresponded to mining sector? Processed foods is not be classified as Agriculture, because they need more electricity than agricultural products such as rice etc. There are unclear sector correspondences.

Response: The uncertainties associated with the missing provincial statistical data has been calculated in our revised manuscript. Please see the response for comment (12).

For the mapping process, as pointed out by the referee, the sectors used for mapping in the household consumption statistics were relatively rough. Actually, in the original statistical data, some provinces (e.g. Fujian) have rough sectors as shown in the old mapping table, while some provinces (e.g. Beijing) and the national average data have more detailed sectors, as shown in the new mapping table below. We used to aggregate the more detailed sectors into the rough

sectors and then map them with the sectors in the MRIO, which might cause sector misclassification. In the revised manuscript, we tried to improve the mapping process by allocating the rough sectors from certain provinces to the more detailed sectors using national average data as proxy before mapping. There were also some translation issues in the old table since the table was originally available in Chinese, which has also been corrected. The new mapping table is shown below, which we believe are more reasonable than the old one. This mapping process is also similar to that used in Wiedenhofer et al. (2017).

For the coal mining sector mentioned by the referee, it is part of the MRIO table that cannot be removed from the analyses. Otherwise the MRIO table will not be balanced. However, the contribution of coal mining sector is really small – It is only 0.5% and 0.1% in rural and urban consumptions respectively. And the deaths related to coal mining is 0.0% and 0.0% for rural and urban respectively.

In the new mapping table, *Food* has more detailed categories and has been mapped to different sectors in MRIO. For example, grain, starches and tubers, beans and bean products, meats, poultry, egg, aquatic products and related products, vegetables; dried and fresh melons and fruits are linked to *Agriculture* in MRIO. Oil and fats, condiments, sugar, tobacco, liquor and beverages, cake, milk and its products, other food are now linked to *Food processing and tobaccos* in MRIO. Dining out is linked to *Hotel and restaurant* in the MRIO.

ID	Sectors in MRIO	Sectors in household consumption data
1	Agriculture	Grain; Starches and Tubers; Beans and Bean Products; Meats, Poultry, Egg, Aquatic Products and Related Products; Vegetables; Dried and Fresh Melons and Fruits
2	Coal mining	Residence-Water, Electricity and Fuels
3	Petroleum and gas	Residence-Water, Electricity and Fuels
4	Metal mining	Residence-House
5	Nonmetal mining	Residence-House
6	Food processing and tobaccos	Oil and Fats; Condiments; Sugar, Tobacco, Liquor and Beverages; Cake, Milk and Its Products; Other Food
7	Textile	Clothing Materials; Bed Articles
8	Clothing, leather, fur, etc.	Garments; Shoes; Other Clothing
9	Wood processing and furnishing	Furniture; Furniture Materials
10	Paper making, printing, stationery, etc.	Cultural and Recreational Articles; Teaching material
11	Petroleum refining, coking, etc.	Transport equipment fuels and parts
12	Chemical industry	Medicine; Health Products
13	Nonmetal products	Residence-House

14	Metallurgy	Residence-House
15	Metal products	Household Appliances
16	General and specialist machinery	Household Appliances
17	Transport equipment	Transportation Facility
18	Electrical equipment	Household Appliances
19	Electronic equipment	Communication Facility
20	Instrument and meter	Medical Appliances; Health Care Appliances
21	Other manufacturing	Room Decorations; Household Articles for Daily Use; Miscellaneous Goods
22	Electricity and hot water production and supply	Residence-Water, Electricity and Fuels
23	Gas and water production and supply	Residence-Water, Electricity and Fuels
24	Construction	Residence-House
25	Transport and storage	Traffic Fare
26	Wholesale and retailing	Expenditure on Food (except Dining Out), clothing, Household Appliances, Transport and Communications Facility, Cultural and Recreational Articles, and Miscellaneous Goods
27	Hotel and restaurant	Dining Out
28	Leasing and commercial services	Miscellaneous Services
29	Scientific research	Education Tuition
30	Other services	Food Processing Service Fees; Tailoring and Laundering Services; Housing Services; Household Services; Health Care Services; Transport equipment using and upkeep fare; Communication Services; Expenditure of Culture and Recreation; Education Tuition; Miscellaneous Services

(15) Page 7, Lines 216-217: The two equations captures that household consumptions of income class i of province s affected $PM_{2.5}$ emissions of other provinces indirectly. Again, how the emissions data were converted into the $PM_{2.5}$ -related premature deaths? There is no mathematical explanation. The subscript t is the final demand category that you have already defined above.

Response: These two equations are used to calculate the emissions embodied in each income group. The conversion between emissions and $PM_{2.5}$ -related premature deaths are achieved through the GEOS-Chem adjoint model, which is described in the following section (S5).

(16) Page 7, L228: What is the correlation equation? What is the database on the statistical analysis of the relationship between biomass consumption and income per capita? This equation is important in determining the direct emissions from each household income class.

Response: For rural biomass consumption emissions, we use a correlation equation between biomass consumption and income per capita adopted from Peng et al. to allocate the emissions into various rural income groups. This correlation equation is fitted using hierarchical regression based on the survey-based per capita income and biofuel consumption, which is shown below:

$$t_{bio,j} = 0.7072 \times \alpha_j^{-0.18}$$

where α_j is the per capita income of income group j ; $t_{bio,j}$ is the biomass consumption for people at the income group j . Usually, the poor households tend to consume more biomass and less commercial fuel²³. More details about this correlation equation can be found in Peng et al. (2019).

References:

Peng, L. *et al.* Underreported coal in statistics: A survey-based solid fuel consumption and emission inventory for the rural residential sector in China. *Applied Energy* **235**, 1169-1182 (2019).

(17) Page 8, L253: This is the connection between the footprint analysis and the health impact analysis. It seems to me that Eq. (23) is a 'linear' impact model. My understanding is that the non-linear impact model of Eq. (1) can be used to convert the PM_{2.5} emissions into the health impacts, nevertheless the authors did not use the non-linear impact model based on the emissions embedded in household consumptions of each income class of each province. The benchmark health risk coefficients were estimated by using the satellite-derived PM_{2.5} concentrations and then the health risk coefficients were 'proportionally' used to estimate the health impacts embedded in household consumptions of each income class of each province. The marginal health impacts are exactly the same over the emitted PM_{2.5} emissions? If my understanding is correct, this model assumption is really correct? Smaller (larger) additional emissions may induce smaller (larger) additional health impacts.

Response: Eq. 23 (now Eq. 21) is the combination of two linear equations that have been validated and used in previous studies (Chafe et al., 2014; Turner et al., 2015; Zhang et al., 2015; Zhang et al., 2017).

The first one is using the semi-normalized sensitivities to represent the contributions of precursor emissions from a certain source to the total PM_{2.5} concentrations. The GEOS-Chem adjoint model offers a computationally efficient approach to calculate the sensitivity of the cost function to emissions in all model grid cells in a single model run. It has considered the physical process and chemical reactions inside the model. Then we could sum the adjoint sensitivities across different dimensions from a certain source and interpret them as percentage contributions to the total sensitivity, which is the contribution of this source. The method has been used in previous source apportionment studies (Turner et al., 2015; Zhang et al., 2015).

The second one is the direct proportion approach, which assumes a linear relationship between the proportion of total PM_{2.5} concentration to the proportion of total mortality. The scientific basis of this assumption has been validated by a GBD research, GBD MAPS (GBD MAPS Working Group, 2016). In general, due to the nonlinearity of the IER functions, the source-specific contribution to PM_{2.5} mortality is non-uniform along the curve. Each source contributes a certain proportion to total PM_{2.5} concentration, but we could not assume the contribution is

located in distinctive intervals of concentration. The direct proportion approach is insensitive to the order in which each source is removed from the total concentration. It also has the advantage that the sum of mortality estimates from all sources equals the mortalities from ambient PM_{2.5} exposure. Previous studies have used the direct proportion approach to solve the non-linear problem (Chafe et al., 2014; Zhang et al., 2017).

References:

Chafe, Z. A. *et al.* Household cooking with solid fuels contributes to ambient PM_{2.5} air pollution and the burden of disease. *Environ. Health Perspect.* **122**, 1314-1320 (2014).

GBD MAPS Working Group. Burden of Disease Attributable to Coal-Burning and Other Major Sources of Air Pollution in China. Special Report 20. Boston, MA:Health Effects Institute (2016).

Turner, M. D. *et al.* Premature deaths attributed to source-specific BC emissions in six urban US regions. *Environmental Research Letters* **10**, 114014 (2015).

Zhang, L. *et al.* Source attribution of particulate matter pollution over North China with the adjoint method. *Environmental Research Letters* **10**, 084011 (2015).

Zhang, Q. *et al.* Transboundary health impacts of transported global air pollution and international trade. *Nature* **543**, 705-709 (2017).

(18) Pages 8-9, S6: This analysis is potentially interesting, however I am wondering whether or not the estimated results are reliable and robust empirically.

Response: We have fully discussed the uncertainties within each step of our work and provided an overall estimate of the total errors in the revised manuscript. Below is the detailed discussion. The estimation of premature deaths associated with household consumption related atmospheric pollutant emissions are subject to a number of uncertainties, due to the limitation and assumption inherit of the models used in each step. The uncertainty ranges (95% confidence interval) in different steps of our analysis and the overall uncertainties are discussed below.

First, there are uncertainties in the air pollutant emission inventory, due to the incomplete knowledge on activity level, combustion/production technology and emission factors. Zhao et al. (2011) estimate that the uncertainties of China's anthropogenic SO₂, NO_x, PM_{2.5}, BC, and OC emissions are -14%~13%, -13%~37%, -17%~54%, -25%~136% and -40%~121%, respectively. The MEIC model used in this study have similar uncertainty ranges, with lower uncertainties associated with SO₂ and NO_x than BC and OC. In addition, the MEIC model has been widely used and proved reliable in chemical transport simulations when validated against surface and satellite observations (Li et al., 2015; Zheng et al., 2015; Hu et al., 2016).

Second, using MRIO analysis and statistical data to allocate production-based emissions to direct and indirect household consumption in different income groups introduces additional uncertainties. Uncertainty in MRIO model is from its data source and data manipulation process, such as sector and region aggregation, data harmonization (Wiedmann et al., 2011; Tukker et al., 2013). By using Monte Carlo simulation, Lin et al. (2014) estimates that the uncertainties of China export-related pollutant emissions is $\pm 50\%$ and the input-output model contributed to ~10% of the total errors. We consider a 10% of uncertainty to represent the errors brought by MRIO analysis. The uncertainties associated with the statistical data mainly come from the

missing of income groups and expenditure patterns data in some provinces, which have been discussed in S3. We use the CV values of each income groups as the uncertainty estimates of the statistical data, as shown in Supplementary Table 4.

Third, errors in the total premature deaths related to PM_{2.5} pollution come from satellite-based PM_{2.5} estimates and health impact models. The PM_{2.5} concentration used here has been calibrated by satellite-based and surface observations, so its uncertainty is relative small, about $\pm 5\%$ on average according to GBD 2013 (Brauer et al., 2016). For the IER model, it is fit by incorporating information on risk due to various emission sources (ambient air pollution, first- and second- hand tobacco smoking, and household indoor air pollution) covering a wide range of PM_{2.5} concentration, but with limited information on actual exposure to ambient PM_{2.5} at higher and lower concentrations (Burnett et al. 2014). Moreover, the IER function is limited to the assumptions that each health endpoint is independent of exposure period, PM_{2.5} composition and toxicity for particles from different sources (Burnett et al. 2014). The uncertainty associated with IER model is quantified by 1,000 simulations using 1,000 sets of parameters provided by Burnett et al. (2014). Uncertainty in the PM_{2.5} data is minor compared to that of IER model and is ignored in the simulations.

Fourth, uncertainties in linking the total premature deaths into each income group also arise from the simulation of GEOS-Chem and its adjoint. The model simulations can contain a lot of uncertainties due to the uncertainty in emission inputs and the model's imperfect representation of chemical and physical process, such as the chemical conversion and physical transport and transform. For GEOS-Chem and its adjoint, the uncertainties come from both the forward simulation and the backward response process. For the forward process, we conduct a comparison between the modeled and the satellite-derived PM_{2.5} concentration among seven regions in China (Supplementary Figure 5), and the R range from 0.28 to 0.95. For the backward process, the calculation is more complex due to its frequent integration with the forward process. Moreover, the source attribution using adjoint sensitives implicitly neglect the nonlinear response of PM_{2.5} to emissions changes. Due to the complex interaction process, there are very few statistical quantification for the uncertainties in GEOS-Chem adjoint simulation so far. We used 30% to represent its uncertainty in our study.

The overall uncertainties involved in the regional household consumption related premature deaths attributable to PM_{2.5} in different income groups are determined by uncertainties in total PM_{2.5} related death calculated by IER functions and the fractional contribution of each income group calculated using GEOS-Chem adjoint model, emission inventory, MRIO model and the statistical data. Uncertainty of total PM_{2.5} related death follows the distribution generated by 1,000 sets of IER parameters. Uncertainties in the fractional contributions are addition in quadrature of errors in GEOS-Chem adjoint model, emissions inventory, MRIO model and the statistical data. Finally, the overall uncertainties are derived from aggregations of errors above. We present the 95% confidence interval of the overall uncertainties in our study.

References:

Brauer, M. *et al.* Ambient Air Pollution Exposure Estimation for the Global Burden of Disease 2013. *Environ. Sci. Technol.* **50**, 79-88 (2016).

- Burnett, R. T. *et al.* An Integrated Risk Function for Estimating the Global Burden of Disease Attributable to Ambient Fine Particulate Matter Exposure. *Environ. Health Perspect.* **122**, 397-403 (2014).
- Hu, J., Chen, J., Ying, Q. & Zhang, H. One-year simulation of ozone and particulate matter in China using WRF/CMAQ modeling system. *Atmos. Chem. Phys.* **16**, 10333-10350 (2016).
- Wiedmann, T., Wilting, H. C., Lenzen, M., Lutter, S. & Palm, V. Quo Vadis MRIO? Methodological, data and institutional requirements for multi-region input–output analysis. *Ecol. Econ.* **70**, 1937-1945 (2011).
- Li, X. *et al.* Source contributions of urban PM_{2.5} in the Beijing–Tianjin–Hebei region: Changes between 2006 and 2013 and relative impacts of emissions and meteorology. *Atmos. Environ.* **123, Part A**, 229-239 (2015).
- Tukker, A. & Dietzenbacher, E. Global multiregional input-output frameworks: an introduction and outlook. *Econ. Syst. Res.* **25**, 1-19, (2013).
- Zhao, Y., Nielsen, C. P., Lei, Y., McElroy, M. B. & Hao, J. Quantifying the uncertainties of a bottom-up emission inventory of anthropogenic atmospheric pollutants in China. *Atmos. Chem. Phys.* **11**, 2295-2308 (2011).
- Zheng, B. *et al.* Heterogeneous chemistry: a mechanism missing in current models to explain secondary inorganic aerosol formation during the January 2013 haze episode in North China. *Atmos. Chem. Phys.* **15**, 2031-2049 (2015).

Referee #3 Comments:

This study examined the inequality of household consumption and air pollution-related deaths in 30 provinces of China. It is interesting and the statistical methods used are appropriate. However, I have one major concern about the uncertainty of the results, for that the methodology of this paper involved multiple models and procedures. Thus, the uncertainty should be fully considered and discussed. However, this important information is lacked in the result and discussion sections.

Response: We thank the referee for the encouragement and the valuable comments to improve our manuscript. Now, we have discussed the uncertainties associated with each model and step, and then provided the overall uncertainties in the figures, tables and the main text. Below is the detailed discussion.

The estimation of premature deaths associated with household consumption related atmospheric pollutant emissions are subject to a number of uncertainties, due to the limitation and assumption inherit of the models used in each step. The uncertainty ranges (95% confidence interval) in different steps of our analysis and the overall uncertainties are discussed below.

First, there are uncertainties in the air pollutant emission inventory, due to the incomplete knowledge on activity level, combustion/production technology and emission factors. Zhao et al. (2011) estimate that the uncertainties of China's anthropogenic SO₂, NO_x, PM_{2.5}, BC, and OC emissions are -14%~13%, -13%~37%, -17%~54%, -25%~136% and -40%~121%, respectively. The MEIC model used in this study have similar uncertainty ranges, with lower uncertainties associated with SO₂ and NO_x than BC and OC. In addition, the MEIC model has been widely used and proved reliable in chemical transport simulations when validated against surface and satellite observations (Li et al., 2015; Zheng et al., 2015; Hu et al., 2016).

Second, using MRIO analysis and statistical data to allocate production-based emissions to direct and indirect household consumption in different income groups introduces additional uncertainties. Uncertainty in MRIO model is from its data source and data manipulation process, such as sector and region aggregation, data harmonization (Wiedmann et al., 2011; Tukker et al., 2013). By using Monte Carlo simulation, Lin et al. (2014) estimates that the uncertainties of China export-related pollutant emissions is $\pm 50\%$ and the input-output model contributed to ~10% of the total errors. We consider a 10% of uncertainty to represent the errors brought by MRIO analysis. The uncertainties associated with the statistical data mainly come from the missing of income groups and expenditure patterns data in some provinces, which have been discussed in S3. We use the CV values of each income groups as the uncertainty estimates of the statistical data, as shown in Supplementary Table 4.

Third, errors in the total premature deaths related to PM_{2.5} pollution come from satellite-based PM_{2.5} estimates and health impact models. The PM_{2.5} concentration used here has been calibrated by satellite-based and surface observations, so its uncertainty is relative small, about $\pm 5\%$ on average according to GBD 2013 (Brauer et al., 2016). For the IER model, it is fit by incorporating information on risk due to various emission sources (ambient air pollution, first- and second- hand tobacco smoking, and household indoor air pollution) covering a wide range of PM_{2.5} concentration, but with limited information on actual exposure to ambient PM_{2.5} at higher and lower concentrations (Burnett et al. 2014). Moreover, the IER function is limited to

the assumptions that each health endpoint is independent of exposure period, PM_{2.5} composition and toxicity for particles from different sources (Burnett et al. 2014). The uncertainty associated with IER model is quantified by 1,000 simulations using 1,000 sets of parameters provided by Burnett et al. (2014). Uncertainty in the PM_{2.5} data is minor compared to that of IER model and is ignored in the simulations.

Fourth, uncertainties in linking the total premature deaths into each income group also arise from the simulation of GEOS-Chem and its adjoint. The model simulations can contain a lot of uncertainties due to the uncertainty in emission inputs and the model's imperfect representation of chemical and physical process, such as the chemical conversion and physical transport and transform. For GEOS-Chem and its adjoint, the uncertainties come from both the forward simulation and the backward response process. For the forward process, we conduct a comparison between the modeled and the satellite-derived PM_{2.5} concentration among seven regions in China (Supplementary Figure 5), and the R range from 0.28 to 0.95. For the backward process, the calculation is more complex due to its frequent integration with the forward process. Moreover, the source attribution using adjoint sensitives implicitly neglect the nonlinear response of PM_{2.5} to emissions changes. Due to the complex interaction process, there are very few statistical quantification for the uncertainties in GEOS-Chem adjoint simulation so far. We used 30% to represent its uncertainty in our study.

The overall uncertainties involved in the regional household consumption related premature deaths attributable to PM_{2.5} in different income groups are determined by uncertainties in total PM_{2.5} related death calculated by IER functions and the fractional contribution of each income group calculated using GEOS-Chem adjoint model, emission inventory, MRIO model and the statistical data. Uncertainty of total PM_{2.5} related death follows the distribution generated by 1,000 sets of IER parameters. Uncertainties in the fractional contributions are addition in quadrature of errors in GEOS-Chem adjoint model, emissions inventory, MRIO model and the statistical data. Finally, the overall uncertainties are derived from aggregations of errors above. We present the 95% confidence interval of the overall uncertainties in our study.

References:

- Brauer, M. *et al.* Ambient Air Pollution Exposure Estimation for the Global Burden of Disease 2013. *Environ. Sci. Technol.* **50**, 79-88 (2016).
- Burnett, R. T. *et al.* An Integrated Risk Function for Estimating the Global Burden of Disease Attributable to Ambient Fine Particulate Matter Exposure. *Environ. Health Perspect.* **122**, 397-403 (2014).
- Hu, J., Chen, J., Ying, Q. & Zhang, H. One-year simulation of ozone and particulate matter in China using WRF/CMAQ modeling system. *Atmos. Chem. Phys.* **16**, 10333-10350 (2016).
- Wiedmann, T., Wilting, H. C., Lenzen, M., Lutter, S. & Palm, V. Quo Vadis MRIO? Methodological, data and institutional requirements for multi-region input-output analysis. *Ecol. Econ.* **70**, 1937-1945 (2011).
- Li, X. *et al.* Source contributions of urban PM_{2.5} in the Beijing–Tianjin–Hebei region: Changes between 2006 and 2013 and relative impacts of emissions and meteorology. *Atmos. Environ.* **123, Part A**, 229-239 (2015).
- Tukker, A. & Dietzenbacher, E. Global multiregional input-output frameworks: an introduction and outlook. *Econ. Syst. Res.* **25**, 1-19, (2013).

Zhao, Y., Nielsen, C. P., Lei, Y., McElroy, M. B. & Hao, J. Quantifying the uncertainties of a bottom-up emission inventory of anthropogenic atmospheric pollutants in China. *Atmos. Chem. Phys.* **11**, 2295-2308 (2011).

Zheng, B. *et al.* Heterogeneous chemistry: a mechanism missing in current models to explain secondary inorganic aerosol formation during the January 2013 haze episode in North China. *Atmos. Chem. Phys.* **15**, 2031-2049 (2015).

Other minor comments:

-Title, “air pollution deaths” should be “air pollution-related deaths”. The authors should revise this problem through the paper.

Response: We have revised our manuscript as suggested.

-Discussion and Conclusions, this paper reported many interesting findings. Policy implications of these results are needed, for that this information is important for the policymaker and governments. And comparison with previous studies on the same topic is also needed.

Response: We have revised our manuscript to provide policy implications, e.g. “Policies that promote clean energy (e.g., natural gas and electricity use) in rural households could provide a perfect solution, however, the high prices, lack of accessibilities to natural gas, and traditional consumption behavior might hinder the promotion of such policy³⁸. Because urban residents also suffer the pollution from rural emissions and have higher willingness to pay for alleviating pollution, providing price subsidy within a certain time period might be a possible solution given that urban residents payed more taxes than rural residents. The price of clean energy will be eventually accepted by rural residents with economy developed and income increased.” “Our work provides unprecedented quantitative insight into the supply chain patterns that link final consumption to air-pollution related premature deaths. For example, by tracking the health impact along the supply chains, we show that the cross-regional health impacts of emissions embodied in household consumption of goods and services are much greater than the effects of atmospheric transport across regional boundaries. Moreover, we highlight systematic differences in the impacts of household consumption on residents according to their income-level and location. These findings point to targeted opportunities for pollution abatement, such as direct emissions from solid fuels burned by rural households, but more importantly offer a basis for clean air policies that avoid and redress socio-economic and regional inequities. Reducing emissions throughout the supply chains will require some combination of improved air pollution control technologies, changes in energy mix, and changes in the location of manufacturing⁴⁰. To the extent these changes must be undertaken by less economically developed regions and households, consumption-based policies may better support the needed technology transfer and capital investment while at the same time encouraging more sustainable consumption behaviors.”

We also compared our results with previous studies on the same topic: “Our results indicate that income and thus the scale of household expenditures are closely related to the air pollution-related deaths related to the consumption of urban households in China. For example, we estimate that 10,000 very rich urban consumers account for 5.4 premature air pollution-related

deaths (95% CI: 3.81–7.0) per year—a factor of 3.3 times more than 10,000 extremely poor urban consumers (1.6 deaths, 95% CI: 1.2–2.1). This is within the line of other studies related to income and environmental impact; impact of rich urban consumers are higher than poor consumers by a factor of 3.8–9.5 for different environmental indicators (i.e., CO₂ emissions, air and water pollutant emissions, and water use)^{18,23,24,38}. Our work provides additional insight to this discussion by adding air pollution-related premature deaths as a new indicator.”

References:

- 18 Cai, B., Liu, B. & Zhang, B. Evolution of Chinese urban household's water footprint. *Journal of Cleaner Production* **208**, 1-10 (2019).
- 23 Zhang, J., Yu, B., Cai, J. & Wei, Y.-M. Impacts of household income change on CO₂ emissions: An empirical analysis of China. *Journal of Cleaner Production* **157**, 190-200 (2017).
- 24 Wiedenhofer, D. *et al.* Unequal household carbon footprints in China. *Nature Climate Change* **7**, 75 (2016).
- 38 Zheng, B. *et al.* Trends in China's anthropogenic emissions since 2010 as the consequence of clean air actions. *Atmos. Chem. Phys.* **18**, 14095-14111 (2018).
- 39 Liu, L.-C. & Wu, G. Relating five bounded environmental problems to China's household consumption in 2011–2015. *Energy* **57**, 427-433 (2013).
- 40 Wang, H. *et al.* Trade-driven relocation of air pollution and health impacts in China. *Nature communications* **8**, 738 (2017).

-Line 191, please provide references after the statement “As concluded by previous studies”.

Response: This paragraph has now been removed from our manuscript.

-Methods, Line 206, the reason of choosing four months (January, April, July and October) of simulations for the total year needs to be well justified.

Response: We chose four months (January, April, July and October) of simulations to save the computation time of the GEOS-Chem adjoint model in our study, because we have to conduct seven groups of simulations for the seven receptor regions. The four months simulations with one month from each season could provide a good estimate of the annual mean value. We compared the four-month averaged PM_{2.5} concentrations with the annual mean PM_{2.5} concentrations of the model simulations, as shown below. The four-month mean data and the annual mean data have very similar performance when comparing with the same annual mean satellite-based PM_{2.5} data, which indicates that the averaged value from these four months could represent the annual level.

-Methods, Line 204, please provide the reason why Tibet, Hong Kong and Macao were not included in this paper.

Response: It is because the multi-regional input-output (MRIO) model of China used in this study to attribute provincial emissions to household consumption only includes 30 provincial-level administrative divisions (Tibet, Macao, Hong Kong and Taiwan are not included) and 30 aggregated sectors.

-Supplementary materials, Table S4, provide the results of 95% confidence interval for the consumption related deaths and consumption related deaths per capita.

Response: 95% confidence interval added.

-Supplementary materials, Figure S1, revise “Ya ngtze” to “Yangtze”.

Response: Revised.

Some language errors. The authors should check this problem throughout the manuscript.

Response: We have carefully read through the manuscript and revised the language mistakes.

-Line 93, “from poorest on the left to richest on the right” should be “from the poorest on the left to the richest on the right”.

Response: Revised.

-Line 102, “greater those” should be “greater than those”.

Response: Revised.

Referee #4 Comments:

This is interesting work with potentially important results. However, a central message is missing, and it is not entirely clear what part of the results agrees/disagrees with previous studies, and what part is novel and remarkable enough to justify a high-profile publication. Somehow, the article needs a “punch-line”, and more pronounced indications of policy implications. I recommend that publication in Nature Comm. is considered after major revision of the manuscript.

Response: We thank the referee for the constructive comments. We have revised our manuscript extensively to emphasize more strongly the key insight of our work and also compared our results with previous studies. Please see the revised *Discussion and Conclusions* section.

Major comments:

The calculations have been performed using IER functions of Burnett et al. (ref. 19), indicating that this is the latest model (supplement 1.66). This is not correct, as the latest global burden of disease evaluation uses updates by Cohen et al. (2017), yielding about 35% higher mortality estimates compared to previous work. While I do not suggest repeating all calculations with the new IERs, it will be needed to discuss this and indicate that present estimates are lower limits.

Response: We thank the referee for pointing out this issue. We have revised our words and added discussion in the *Supplementary Information*.

“The IER model is developed by Burnett et al. (2015), and has been used to estimate the PM_{2.5} related premature deaths in previous studies (Lee et al., 2015; Lelieveld et al., 2015; Zhang et al., 2017). In Cohen et al. (2017), an updated version of the function are provided, yielding about 35% higher mortality estimates compared to previous work. Therefore, the results in our study present the lower limits of the estimates.”

References:

Burnett, R. T. *et al.* An Integrated Risk Function for Estimating the Global Burden of Disease Attributable to Ambient Fine Particulate Matter Exposure. *Environ. Health Perspect.* 122, 397-403 (2014).

Cohen, A. J. *et al.* Estimates and 25-year trends of the global burden of disease attributable to ambient air pollution: an analysis of data from the Global Burden of Diseases Study 2015. *The Lancet* **389**, 1907-1918 (2017).

Lee, C. J. *et al.* Response of global particulate-matter-related mortality to changes in local precursor emissions. *Environ. Sci. Technol.* **49**, 4335-4344 (2015).

Lelieveld, J., Evans, J. S., Fnais, M., Giannadaki, D. & Pozzer, A. The contribution of outdoor air pollution sources to premature mortality on a global scale. *Nature* **525**, 367-371 (2015).

Zhang, Q. *et al.* Transboundary health impacts of transported global air pollution and international trade. *Nature* **543**, 705-709 (2017).

An important result is that a similar number of deaths were caused by rural and urban consumption, and that residential energy use` is a leading cause. I believe that this aspect could

be highlighted more strongly, as discussions on air pollution typically focus on the urban environment and the role of traffic.

Response: We thank the referee for the valuable comments. In the revised manuscript, we updated the GEOS-Chem adjoint model according to the suggestions by the GEOS-Chem support team (http://wiki.seas.harvard.edu/geos-chem/index.php/Particulate_matter_in_GEOS-Chem) and had higher estimation of organic carbon concentrations, resulting in higher contribution of rural direct emissions to the air pollution related deaths. As a result, the number of deaths attributed to rural consumption is now higher than that related to urban consumption. We have revised our manuscript to emphasize this finding.

Results section: “On average, the number of deaths attributed to rural consumption is higher than that related to urban consumption when considering both direct and indirect emissions.”

Discussion and Conclusions section: “Although substantial contribution of solid fuel use on air pollution in China has been investigated³⁵⁻³⁷, we find unexpected higher contribution of rural household consumption to air pollution-related deaths in China. These findings further emphasize the great importance of mitigating emissions from direct emissions of rural households, given that current policies focus more on urban pollution.”

References:

35 Zhang, J. & Smith, K. R. Household air pollution from coal and biomass fuels in China: measurements, health impacts, and interventions. *Environmental health perspectives* **115**, 848-855 (2007).

36 Liu, J. *et al.* Air pollutant emissions from Chinese households: A major and underappreciated ambient pollution source. *Proceedings of the National Academy of Sciences* **113**, 7756-7761 (2016).

37 Zhao, B. *et al.* Change in household fuels dominates the decrease in PM_{2.5} exposure and premature mortality in China in 2005–2015. *Proceedings of the National Academy of Sciences* **115**, 12401-12406 (2018).

On the other hand, the result that the average income of those dying prematurely is often lower than that of the consumers causing their deaths is not remarkable.

Response: Yes, previous studies have examined the relationship between income and other environmental impact. They found that the impact of rich urban consumers are higher than poor consumers by a factor of 3.8–9.5 for different environmental indicators (i.e., CO₂ emissions, air and water pollutant emissions, and water use)^{18,23,24,39}. But no previous studies have quantified air pollution-related deaths embodied in household consumption, and our work provides additional insight to this discussion by adding air pollution-related premature deaths as a new indicator.

References:

18 Cai, B., Liu, B. & Zhang, B. Evolution of Chinese urban household's water footprint. *Journal of Cleaner Production* **208**, 1-10 (2019).

23 Zhang, J., Yu, B., Cai, J. & Wei, Y.-M. Impacts of household income change on CO₂ emissions: An empirical analysis of China. *Journal of Cleaner Production* **157**, 190-200 (2017).

24 Wiedenhofer, D. *et al.* Unequal household carbon footprints in China. *Nature Climate Change* **7**, 75 (2016).

39 Liu, L.-C. & Wu, G. Relating five bounded environmental problems to China's household consumption in 2011–2015. *Energy* **57**, 427–433 (2013).

The conclusion that “reducing solid fuel consumption by rural households and encouraging sustainable consumption in urban areas may be the most effective means of reducing the environmental and health inequalities related to Chinese air pollution” appears relevant. However, the part “environmental and health inequalities” could be replaced by “health consequences”. On the other hand, is this really new? The last section writes “substantial reductions in air pollution deaths are possible by reducing the combustion of household solid fuel in China”, which is not surprising.

Response: Yes, previous studies have investigated and reported the substantial contribution of solid fuel use on air pollution in China³⁵⁻³⁷. However, they only focused on the direct impact of household fuel use, without considering the indirect impact of household consumption through supply chain. For the first time, our study developed the quantitative relationship between household consumption and air pollution-related premature deaths from both direct and indirect perspective. We find unexpected higher contribution of rural household consumption to air pollution-related deaths in China, which further emphasize the great importance of mitigating emissions from direct emissions of rural households, given that current policies focus more on urban pollution. But more importantly, our study offers a basis for clean air policies that avoid and redress socio-economic and regional inequities.

We now revised the related discussion on direct fuel consumption as follows:

“Although substantial contribution of solid fuel use on air pollution in China has been investigated³⁵⁻³⁷, we find unexpected higher contribution of rural household consumption to air pollution-related deaths in China. These findings further emphasize the great importance of mitigating emissions from direct emissions of rural households, given that current policies focus more on urban pollution. Indeed, our results likely underestimate air pollution-related deaths by rural households because neglecting the impact of indoor air pollution from solid fuel use. Policies that promote clean energy (e.g., natural gas and electricity use) in rural households could provide a perfect solution, however, the high prices, lack of accessibilities to natural gas, and traditional consumption behavior might hinder the promotion of such policy³⁸. Because urban residents also suffer the pollution from rural emissions and have higher willingness to pay for alleviating pollution, providing price subsidy within a certain time period might be a possible solution given that urban residents pay more taxes than rural residents. The price of clean energy will be eventually accepted by rural residents with economy developed and income increased.”

References:

35 Zhang, J. & Smith, K. R. Household air pollution from coal and biomass fuels in China: measurements, health impacts, and interventions. *Environmental health perspectives* **115**, 848–855 (2007).

36 Liu, J. *et al.* Air pollutant emissions from Chinese households: A major and underappreciated ambient pollution source. *Proceedings of the National Academy of Sciences* **113**, 7756-7761 (2016).

37 Zhao, B. *et al.* Change in household fuels dominates the decrease in PM_{2.5} exposure and premature mortality in China in 2005–2015. *Proceedings of the National Academy of Sciences* **115**, 12401-12406 (2018).

38 Zheng, B. *et al.* Trends in China's anthropogenic emissions since 2010 as the consequence of clean air actions. *Atmos. Chem. Phys.* **18**, 14095-14111 (2018).

The last paragraph (l.173-181) seems trivial.

Response: We have now replaced the last paragraph by the following discussion to emphasize our new findings and their policy implications: “Our work provides unprecedented quantitative insight into the supply chain patterns that link final consumption to air-pollution related premature deaths. For example, by tracking the health impact along the supply chains, we show that the cross-regional health impacts of emissions embodied in household consumption of goods and services are much greater than the effects of atmospheric transport across regional boundaries. Moreover, we highlight systematic differences in the impacts of household consumption on residents according to their income-level and location. These findings point to targeted opportunities for pollution abatement, such as direct emissions from solid fuels burned by rural households, but more importantly offer a basis for clean air policies that avoid and redress socio-economic and regional inequities. Reducing emissions throughout the supply chains will require some combination of improved air pollution control technologies, changes in energy mix, and changes in the location of manufacturing⁴⁰. To the extent these changes must be undertaken by less economically developed regions and households, consumption-based policies may better support the needed technology transfer and capital investment while at the same time encouraging more sustainable consumption behaviors.”

References:

40 Wang, H. *et al.* Trade-driven relocation of air pollution and health impacts in China. *Nature communications* **8**, 738 (2017).

Minor comments:

Although the text generally reads well, a few language mistakes (grammar) need to be corrected.

Response: We have carefully read through the manuscript and revised the language mistakes.

l.89: Please define “indirect” emissions more clearly, e.g. in the caption of fig. 1.

Response: We have revised our manuscript to better clarify the definition of “indirect emissions”: “...28% (305 thousand; 95% CI: 207–402) are related to emissions embodied in household consumption of goods and services (refer to indirect emissions hereafter).”

Reviewers' comments:

Reviewer #1 (Remarks to the Author):

I would like to thank the authors for attending to my comments so diligently. I have no further comments.

Reviewer #2 (Remarks to the Author):

The paper is well written and almost ready for publication.

Finally, I would suggest adding the following references and relating them to your study, then clarify the paper's contribution.

Nagashima, F., Critical structural paths of residential PM2.5 emissions within the Chinese provinces, *Energy Economics* 70, 465-471 2018.

Nagashima, F., Kagawa, S., Suh, S., Nansai, K., Moran, D., Identifying Critical Supply Chain Paths and Key Sectors for Mitigating Primary Carbonaceous PM2.5 Mortality in Asia, *Economic Systems Research* 29, 105-123, 2017.

Reviewer #3 (Remarks to the Author):

The authors have addressed most of my comments. However, I still have some minor comments.

- Mass of data were used in this study. It will be better to provide a brief description on the source and the period of the data at the beginning of the method section.

-The limitations of this study should be carefully discussed in the discussion section. For example, the concentration-response relationship between air pollution and health in IER model from GBD 2010 was mainly based on the studies from Europe and USA. However, this concentration-response association of Chinese population may be different from those in Western developed countries, as the varied chemical component of air pollution, healthcare status and population adaption in these countries. In addition, even though some model uncertainty has been carefully controlled in revised paper, this issue still needs to be reasonable discussed, as too complicate integrated model was used in this study.

-Figure 2 and Figure 4, please explain what the black line denotes in each bar.

Referee #1 Comments:

I would like to thank the authors for attending to my comments so diligently. I have no further comments.

Response: We thank the careful review from the referee and we appreciate that the referee is satisfied with our revisions.

Referee #2 Comments:

The paper is well written and almost ready for publication.

Finally, I would suggest adding the following references and relating them to your study, then clarify the paper's contribution.

Nagashima, F., Critical structural paths of residential PM_{2.5} emissions within the Chinese provinces, *Energy Economics* 70, 465-471 2018.

Nagashima, F., Kagawa, S., Suh, S., Nansai, K., Moran, D., Identifying Critical Supply Chain Paths and Key Sectors for Mitigating Primary Carbonaceous PM_{2.5} Mortality in Asia, *Economic Systems Research* 29, 105-123, 2017.

Response: We appreciate that the referee is satisfied with our revisions. We have added these two references in our revised manuscript as suggested.

Referee #3 Comments:

The authors have addressed most of my comments. However, I still have some minor comments.

Response: We thank the careful review from the referee and we appreciate that the referee is satisfied with our revisions. We also addressed the remaining comments in the revised manuscript.

- Mass of data were used in this study. It will be better to provide a brief description on the source and the period of the data at the beginning of the method section.

Response: We thank the referee for the constructive comments. We now added a summary of the data we used at the beginning of the method section.

-The limitations of this study should be carefully discussed in the discussion section. For example, the concentration-response relationship between air pollution and health in IER model from GBD 2010 was mainly based on the studies from Europe and USA. However, this concentration-response association of Chinese population may be different from those in Western developed countries, as the varied chemical component of air pollution, healthcare

status and population adaption in these countries. In addition, even though some model uncertainty has been carefully controlled in revised paper, this issue still needs to be reasonable discussed, as too complicate integrated model was used in this study.

Response: We thank the referee for the constructive comments. We have a detailed discussion of step-by-step and overall uncertainty in *Supplementary Information*. We now added discussion of limitation and uncertainties in *Discussion and Conclusions*:

Our study was subject to a number of uncertainties and limitations from the use of multiple datasets and complex models. A detailed, quantitative uncertainty analysis for each step of this study was conducted and presented in *Supplementary Information* and the overall uncertainty ranges (95%CI) associated with mortality estimates were presented in Fig. 2 and Fig. 4. First, bottom-up emission inventories are uncertain due to lack of complete data of activity rates and local-measured emission factors⁴². The MEIC emission inventory used in this study has been widely applied in chemical transport models and validated against observations^{43,44}. Second, incomplete income and expenditure data at provincial level contributed to the uncertainties in estimating emissions consumed by each income group. Improvement of statistics reporting system or conducting filed surveys could remedy this situation in the future. Third, sensitivities of emissions to PM_{2.5} exposures were simulated from the GEOS-Chem model and its adjoint, which are also subject to uncertainties due to incomplete knowledge of chemical and physical processes. We compared the modeled and the satellite-derived PM_{2.5} concentration and reasonable correlations were found for most regions in China (Supplementary Figure 7). Last but not least, the IER function used in mortality estimates were developed based on cohort studies in western countries and may introduce additional uncertainties when applying for China due to differences in PM_{2.5} toxicities, population adaption, and healthcare levels. Using concentration-response relationships developed from local cohort studies could improve the estimates of premature mortalities in the future.

-Figure 2 and Figure 4, please explain what the black line denotes in each bar.

Response: The black lines in each bars are the error bars that represent the uncertainty ranges (95%CI) of the estimates. We now added explanations of these black lines in the figure captions of Figure 2 and 4.

REVIEWERS' COMMENTS:

Reviewer #3 (Remarks to the Author):

My concern and comments have been carefully addressed by the authors. I think the paper is almost ready for publication.

Referee #3 Comments:

My concern and comments have been carefully addressed by the authors. I think the paper is almost ready for publication.

Response: We appreciate that the referee is satisfied with our revisions and we thank for all help for improving our manuscript.